# VaPOrS v1.0.1: An automated model for functional group detection and property prediction of organic compounds via SMILES notation

Mojtaba Bezaatpour[1,a], Miikka Dal Maso [a], Matti Rissanen[2, a, b]

*[a] Aerosol Physics Laboratory, Tampere University, 33720 Tampere, Finland*

*[b] Department of Chemistry, University of Helsinki, 00014 Helsinki, Finland*

## Abstract

Volatile organic compounds play a significant role in atmospheric chemistry, influencing air quality and climate change. Accurate prediction of their physical properties is essential for understanding their behavior. This paper introduces VaPOrS (**Va**por **P**ressure in **Or**ganics via **S**MILES) as a comprehensive tool designed to process SMILES notation of organic compounds, identify key functional groups, and apply group-contribution methods for property estimation. The core innovation of VaPOrS lies in its self-contained functional group recognition algorithm, which eliminates dependence on external cheminformatics libraries. The current approach enables fully auditable, easily modifiable, and computationally efficient detection of 30 functional groups required by the SIMPOL method. Compared to existing tools, VaPOrS avoids heavy SMILES-to-graph conversions and can obviate interface overhead, providing orders-of-magnitude speedups for large-scale atmospheric modeling scenarios. While this first implementation focuses on the SIMPOL method for estimating saturation vapor pressure and enthalpy of vaporization, the framework is readily extendable to other group-contribution schemes and thermodynamic properties (e.g., partition coefficients, volatility basis set models, solubility, Henry's law constants). The tool has been validated against manually counted functional groups and experimental saturation vapor pressure data for a diverse set of compounds. Results demonstrate excellent agreement with both the original SIMPOL model and experimental observations, while comparisons with existing tools highlight the robustness and accuracy of the new parsing functions. VaPOrS thus provides a generalizable and computationally efficient platform for property prediction of large molecular datasets,

[1] Corresponding author email address: mojtaba.bezaatpour@tuni.fi (M. Bezaatpour)

[2] Corresponding author email address: matti.rissanen@tuni.fi (M. Rissanen)

facilitating integration into chemical transport and climate models and streamlining the analysis of thousands of organic compounds in atmospheric science applications.

**Keywords:** Vapor pressure, Volatile organic compounds, VaPOrS, SMILES notation, Functional group detection, Atmospheric chemistry, Secondary organic aerosol.

## 1. Introduction

Volatile organic compounds (VOCs) are a diverse group of organic chemicals that significantly impact atmospheric chemistry. Their volatility allows them to easily enter the atmosphere, where they participate in complex chemical reactions that influence air quality, climate, and human health (Mellouki et al. 2015). VOCs are key precursors to both ground-level ozone and secondary organic aerosols (SOA). Ground-level ozone, a harmful air pollutant, forms through the photochemical reactions of VOCs with nitrogen oxides (NOx) in the presence of sunlight (Atkinson 2000). Ozone is a major component of urban smog and poses serious health risks, including respiratory problems and cardiovascular disease (WHO 2005). Secondary organic aerosols, on the other hand, result from the oxidation of VOCs, leading to the formation of particulate matter that can scatter sunlight and affect the Earth's radiative balance (Jimenez et al. 2009). These particles contribute to atmospheric haze, impact visibility, and have been linked to adverse health effects, including lung and heart diseases. Understanding the formation, transformation, and fate of VOCs is thus crucial for predicting their impact on both air quality and climate.

VOCs originate from a variety of sources, both natural and anthropogenic. Natural sources include biogenic emissions from vegetation, such as isoprene and monoterpenes, which are released in large quantities, especially in forested areas (Guenther et al. 1995). Anthropogenic sources are primarily related to industrial activities, transportation, fuel combustion, and the use of solvents and consumer products (Goldstein and Galbally 2007; McDonald et al. 2018). The structural diversity of VOCs poses a challenge for their classification and analysis. They can range from simple hydrocarbons like methane to complex multifunctional molecules containing oxygen, nitrogen, sulfur, and, for example, halogens. This diversity affects their physical properties, reactivity, and environmental behavior, making it necessary to study their properties comprehensively.

The physical properties of VOCs, particularly saturation vapor pressure and enthalpy of vaporization, play a critical role in determining their volatility and phase behavior in the atmosphere. Saturation vapor pressure is a measure of a substance's tendency to evaporate, and it directly influences the distribution of VOCs between the gas and particulate phases (Mattila et al. 1997). Compounds with high saturation vapor pressure are more likely to remain in the gas phase, whereas those with lower saturation vapor pressure can condense onto particulate matter, contributing to SOA formation (Donahue et al. 2006). Enthalpy of vaporization, the energy required to convert a substance from liquid to vapor, is an important property that influences the temperature dependence of saturation vapor pressure. Accurate knowledge of these properties is essential for modeling VOC emissions, their transport, and their transformation in the atmosphere.

The oxidation of VOCs significantly changes their chemical properties. An extreme example is provided by autoxidation, a chain-like oxidation process that propagates by sequential additions of molecular oxygen and resulting peroxy radical rearrangement reactions, which infuses multiple oxygen molecules to the hydrocarbon backbone. This process plummets the saturation vapor pressures of the participating VOC and ultimately forms so-called Highly Oxygenated organic Molecules (HOMs as expressed by Bianchi et al. (Bianchi et al. 2019; Crounse et al. 2013)), crucial intermediates in the formation of SOA. HOM formation is a common property of VOCs and is thus initiated by all common oxidants capable of forming alkyl radicals, including hydroxyl radicals (OH), ozone ($O_3$), and nitrate radicals ($NO_3$) (Zhao et al. 2021; Luo et al. 2023; Rissanen et al. 2014; Berndt et al. 2016) as well as by direct UV photolysis. The resulting HOMs can rapidly condense onto existing particles or sometime even form new particles through nucleation processes, thus contributing to the formation and growth of SOA. The formation of HOMs is heavily influenced by the structure of the precursor VOCs (Ehn et al. 2014). Recent advancements in mass spectrometry and atmospheric simulation techniques have provided valuable insights into the chemical pathways leading to HOMs formation and their role in atmospheric chemistry (Iyer 2023; Bianchi et al. 2019; Vereecken et al. 2018; Jenkin et al. 2019; Iyer et al. 2021a; 2021b). Understanding the volatility of VOCs, and especially HOMs, is therefore essential for improving models of SOA formation.

Traditionally, the determination of saturation vapor pressure and enthalpy of vaporization has relied on experimental methods, such as gas chromatography-mass spectrometry (GC-MS) (Epping and Koch 2023). While these methods provide accurate measurements, they are time-consuming, expensive, and often impractical for large-scale studies involving thousands of

compounds. Moreover, many of the condensable chemicals relevant to atmospheric SOA formation are challenging to measure because their parent molecules are unstable, have never been synthesized, or, importantly, cannot be synthesized due to their inherent instability. These limitations highlight the critical need for approximative methods to estimate the physical properties of these compounds, as direct measurements are often unfeasible.

Given the limitations of experimental approaches, there is an obvious need for computational tools that can predict these properties efficiently. Such tools can complement experimental methods by providing rapid estimates for a wide range of compounds, facilitating large-scale atmospheric modeling studies. Various computational methods have been developed to estimate the saturation vapor pressure of organic compounds, complementing or replacing traditional experimental approaches. Among these, predictive models like COSMO-RS (COnductor-like Screening MOdel for Real Solvents) use quantum chemistry calculations to estimate thermodynamic properties, including saturation vapor pressure. COSMO-RS simulates the solvent environment and molecular interactions, providing a theoretical basis for property estimation. However, its computational intensity can be a drawback for large-scale applications (Klamt 1995; Eckert and Klamt 2002; Klamt et al. 1998). On the other hand, group contribution methods estimate saturation vapor pressure based on the contribution of structural groups within a molecule. These methods leverage empirical correlations derived from extensive datasets but often lack precision for complex or highly functionalized molecules (Joback and Reid 1987; Myrdal and Yalkowsky 1997). The Nannoolal method is a computational approach used to estimate the boiling points and saturation vapor pressures of organic compounds based on their molecular structure. This method utilizes group contribution techniques, where the overall properties of a molecule are determined by summing the contributions of its individual structural groups, such as functional groups and bonding patterns (Nannoolal et al. 2008). The EVAPORATION method is a tool designed for estimating the saturation vapor pressure of organic molecules, accounting for contributions from the carbon skeleton, functional groups, and their interactions, and it adjusts for functionalized diacids with empirical modifications. It predicts the saturation vapor pressure of various compounds using only molecular structure as input, making it applicable to a wide range of molecules (Compernolle et al. 2011). The SIMPOL method, developed by Pankow and Asher (Pankow and Asher 2008), is a widely used group contribution method that correlates the presence of specific functional groups in a molecule to its vapor pressure. By summing the contributions of individual functional groups, the method provides an estimate of the compound's vapor

pressure. The SIMPOL method has been validated against experimental data and is recognized for its accuracy and applicability to a wide range of organic compounds. Importantly, since all these methods employ group contribution techniques, they could potentially be integrated into automated tools, enabling the efficient handling of large datasets and complex compounds with minimal manual effort.

Despite the usefulness of existing tools such as UManSysProp (Topping et al. 2016) for vapor pressure estimation, they show limitations in correctly identifying functional groups in a range of organic molecules. These shortcomings can lead to significant deviations in predicted vapor pressures, particularly for multifunctional species relevant to atmospheric oxidation and SOA formation. Our observations of such discrepancies provided the main motivation for this work. To address these deficiencies, we developed a Python-based computational framework named VaPOrS (**Va**por **P**ressure in **Or**ganics via **S**MILES) to process SMILES (Simplified Molecular Input Line Entry System) notation of VOCs, automatically identify functional groups, and apply group-contribution methods for property estimation.

The core innovation of VaPOrS lies in its self-contained SMILES parsing and group recognition algorithm, which eliminates reliance on external cheminformatics libraries such as OpenBabel. Instead of depending on SMARTS-based pattern matching, VaPOrS explicitly searches for all possible patterns of each functional group directly from the SMILES string, ensuring full control over the detection logic. This approach makes the tool both transparent and adaptable, enabling straightforward extension to new group definitions and methods without external dependencies. In its current version, VaPOrS implements the SIMPOL method by detecting 30 functional groups required for estimating saturation vapor pressure and enthalpy of vaporization. However, this is only the first demonstration of the framework: the same group recognition functions can be applied to other parameterization schemes (e.g., group additivity, volatility basis set (VBS) models, partition coefficients, Henry's law constants), making VaPOrS a general platform for group-contribution modeling rather than a tool restricted to vapor pressure prediction. Therefore, the development of VaPOrS addresses several challenges:

1. Automated and auditable functional group detection: Eliminating manual identification, reducing error potential, and providing full transparency in detection logic.

2. Computational efficiency: By bypassing Pybel's SMILES-to-graph conversions, VaPOrS reduces overhead, enabling much faster execution for large-scale atmospheric simulations involving thousands of compounds across many steps and grid cells.

3. Scalability and flexibility: Capable of processing thousands of SMILES strings within seconds, with design features that make it portable to high-performance computing (HPC) environments and easily translatable to compiled languages such as Fortran for integration into large-scale chemical transport and climate models.

The potential applications of VaPOrS extend well beyond SIMPOL. As an instance, Pichelstorfer et al. (Pichelstorfer et al. 2024) recently introduced the auto-APRAM-fw framework for predicting the molecular structures of highly oxygenated organic molecules (HOMs) generated during autoxidation, outputting results in SMILES notation. Integrating these HOMs into atmospheric models requires both functional group identification and vapor pressure estimation, a process that becomes resource-intensive if performed manually or through non-specialized methods. VaPOrS provides an automated and efficient solution to this challenge by detecting functional groups directly from the SMILES strings and applying parameterizations such as SIMPOL for vapor pressure. More generally, the framework can facilitate integration of large molecular datasets into existing atmospheric databases and simulation platforms, advancing the predictive capability of SOA formation models. Some of the key atmospheric databases and models that could benefit from this tool are discussed in the results and discussion section.

## 2. Methodology

The developed VaPOrS begins by processing the SMILES notation of the target chemical compounds. SMILES strings are used as inputs to identify the molecular components most relevant to volatility predictions, such as number of carbon atoms, functional groups, and cyclic structures. Altogether 30 functional groups parameterized in the SIMPOL method are identified. Several functions are required to efficiently parse and interpret the structure of each compound. The model consists of two main parts. The first part is dedicated to identifying and counting the number of functional groups within the target while the second part focuses on calculating the saturation vapor pressure as a function of temperature and providing the corresponding Antoine equation parameters.

### 2.1. Functional group identification

The first part of the model identifies and counts 30 distinct structural groups in organic compounds that are recognized by the SIMPOL method as influential on saturation vapor pressure. These groups include functional components like carbonyl groups (-C=O), hydroxyl groups (-OH), amines (-NR$_3$), ethers (-R-O-R-), and several others (see Figure 1). The model processes the SMILES string of each compound and applies functions to detect the presence and quantity of these groups. To achieve this, the SMILES string is parsed by several dedicated functions, each corresponding to a specific functional group (e.g., O-atom in a carbonyl or hydroxyl). The results are stored as variables that represent the number of occurrences of each group within the molecule. These counts are then used as input parameters in the saturation vapor pressure calculation. This identification process is critical, as the functional groups detected are the most impactful to the molecule's saturation vapor pressure. By systematically identifying the relevant groups for each compound, the model prepares the necessary data for the next phase of thermodynamic calculations. At the current stage, VaPOrS operates using canonical SMILES as input. This choice was made because canonical SMILES are the most common representation in chemical databases and ensure a unique representation of each molecule, which simplifies functional group detection and validation. Although alternative SMILES forms (e.g., kekulized or non-canonical variations) can represent the same molecule, these are not yet supported in the present version. To maintain consistency, users should therefore provide canonical SMILES when running VaPOrS. The following subsections 2.1.1 to 2.1.19 provide detailed descriptions of the first steps of the model computations.

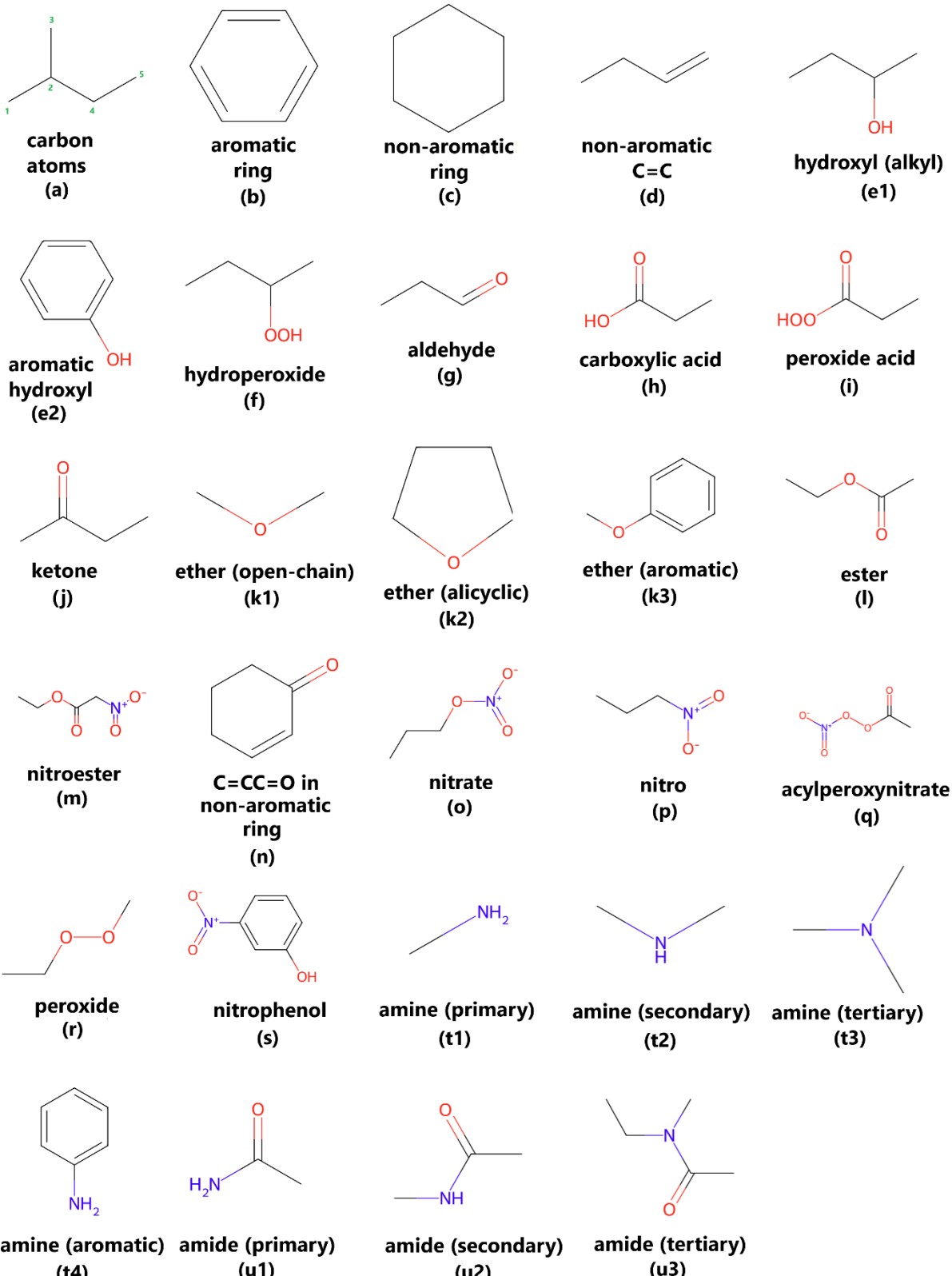

**Figure 1.** Overview of chemical functional groups identified and analyzed by the VaPOrS, including number of

a) carbon atoms, b) aromatic and c) non-aromatic rings, d) non-aromatic double bonds (C=C), e) hydroxyl (alkyl

(e1) and aromatic (e2)), f) hydroperoxide, g) aldehyde, h) carboxylic acid, i) peroxy acid, j) ketone, k) ethers

(open-chain (k1), alicyclic (k2), and aromatic (k3)), l) ester, m) nitroester, n) C=CC=O in non-aromatic rings, o)

nitrate, p) nitro, q) acylperoxynitrate, r) peroxide, s) nitrophenol, t) amines (primary (t1), secondary (t2), tertiary (t3), and aromatic (t4)), and u) amides (primary (u1), secondary (u2), and tertiary (u3)).

### 2.1.1. Carbon atoms

Carbon atoms are an essential component in all organic molecules, and their number forms the basis for the calculations of molecular properties utilizing group additivity. The `carbon_number(s)` function counts the occurrences of both uppercase ('C', representing carbon atoms in alkyl chains) and lowercase ('c', representing carbon atoms in aromatic rings) characters in the SMILES string. The total number of carbon atoms is then calculated by summing these occurrences.

### 2.1.2. Identification of branches and rings in SMILES

Since SMILES notation uses parentheses to denote branching in molecular structures, and numeric indicators to specify ring closure, the model incorporates specialized functions to handle these aspects.

- The function `find_closing_parenthesis(s, open_index)` is designed to locate the corresponding closing parenthesis for a given open parenthesis in the SMILES string. This is crucial for identifying the boundaries of branched substructures. Similarly, `find_opening_parenthesis(s, close_index)` identifies the opening parenthesis corresponding to a closing parenthesis, enabling proper identification of substructures. These functions ensure that the branching parts of the molecules are accurately interpreted, which is essential for correctly assigning functional groups.
- The function `find_cycle_number(s)` is developed to parse through the SMILES string and identify the largest numeric index used for ring closure. This index represents the total number of rings in the compound. For example, if a compound contains three rings, its SMILES notation will include ring closure indicators numbered 1, 2, and 3. Each ring is assigned a specific index sequentially as it is encountered while writing the SMILES. Therefore, the largest numeric index in the SMILES indicates the total number of rings present in the structure.

To illustrate the importance of correctly matching parentheses in the algorithm, consider the following example. The pattern `O=C1C(=C...1...)...` at the beginning of a SMILES notation

may suggest the presence of a C=C–C=O group within a non-aromatic ring (such as in

cyclohex-2-enone, see example (n) in Figure 1).The parentheses must correspond to each other

in the above-mentioned pattern, enclosing the second occurrence of '1'. For instance, in the

SMILES string `O=C1C(=CC)CC(CC1)C`, although it initially appears to match the pattern, the

second '1' does not reside between the matching parentheses. As depicted in Figure 2(a), this

structure places the C=C bond outside the ring, diverging from the characteristics of the

intended functional group. By contrast, in the strings `O=C1C(=CCCC1)C` and

`O=C1C(=C(C)C(C)C)CCC1)C`, the second '1' is properly enclosed within the matching

parentheses (highlighted in bold), adhering to the pattern and satisfying the criteria for

detecting the C=C–C=O group within a non-aromatic ring. Notably, the latter example contains

more carbon atoms, making it easier to count and visually distinguish the structure's

complexity. This highlights the versatility of the algorithm in handling SMILES strings of

varying complexity, even when the carbon atom count increases.

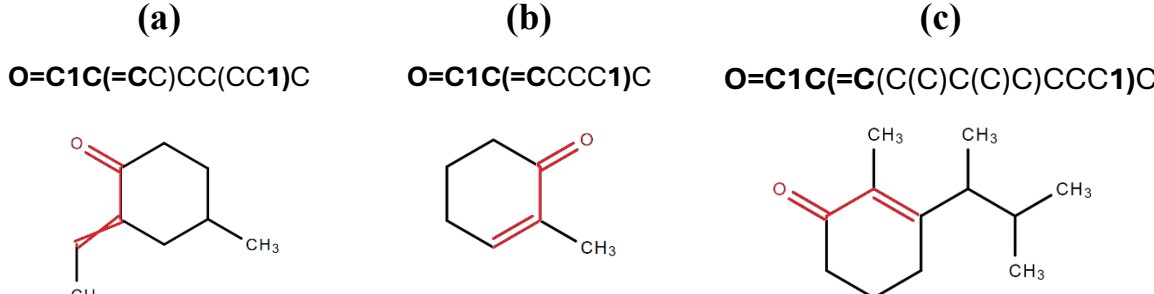

**(a)**         **(b)**         **(c)**

**O=C1C(=CC)CC(CC1)C**    **O=C1C(=CCCC1)C**    **O=C1C(=C(C(C)C(C)C)CCC1)C**

**Figure 2**. Structural representations of SMILES strings demonstrating the importance of

correct parenthesis matching for the identification of C=C–C=O groups in non-aromatic

rings. Structure (a) (`O=C1C(=CC)CC(CC1)C`) incorrectly places the C=C bond outside the ring,

while structures (b) and (c) (`O=C1C(=CCCC1)C` and `O=C1C(=C(C(C)C(C)C)CCC1)C`) adhere to

the correct pattern.

*2.1.3. Aromatic and non-aromatic rings*

Organic molecules often contain cyclic structures, which can either be aromatic (such as

benzene rings) or non-aromatic (such as cyclohexane). The model uses two specific functions,

`aromatic_ring(s)` and `non_aromatic_ring(s)`, to identify and quantify these rings based

on the SMILES notation. The function `aromatic_ring(s)` utilizes the previously described

`find_cycle_number(s)` function to identify the largest ring number within the SMILES

string. This number represents the highest numeric indicator for cyclic structures (e.g., `c1ccccc1` in SMILES denotes a benzene ring with the digit '1' marking the ring). The function then constructs a list of potential aromatic ring representations, such as `'c1'`, `'c2'`, …, `'cn'`, where `'n'` is the index of the last detected ring showing how many rings are in the structure. It then iterates over the SMILES string, counting the occurrences of these aromatic rings. To avoid double-counting, the final count is halved, since each ring closure is represented by two digits (e.g., `c1...c1`). The function returns the total number of aromatic rings present in the molecule.

Similarly, the `non_aromatic_ring(s)` function detects non-aromatic cyclic structures. Like the aromatic ring detection, the function constructs list of possible ring representations, including combinations such as `'C1'`, `'N1'`, `'O1'` and so on, based on the largest ring index. The function returns the total number of non-aromatic rings detected in the molecule.

### 2.1.4. *Non-aromatic double-bonded carbon atoms*

In addition to identifying cyclic structures, the model also detects non-aromatic double bonds between two carbon atoms, a common feature in organic molecules that strongly influences their chemical and physical properties. The function `double_bound_nonaromatic_carbons(s)` is designed to identify occurrences of double-bonded carbon atoms in non-aromatic structures. This function operates by scanning the SMILES string for occurrences of `=C`, which indicates a double bond involving a carbon atom. The following steps are used to ensure accurate detection of non-aromatic double bonds:

- Ring number detection: As with previous functions, `find_cycle_number(s)` is employed to identify the largest numeric ring closure indicator, which is used to distinguish between ring-bound and non-ring-bound carbons.
- Pattern matching: The function scans the SMILES string for `=C`, which denotes a double bond with a carbon atom. Upon finding such occurrences, additional checks are performed to ensure the carbon atoms involved in the double bond are not part of an aromatic system:

  i. Straight chains: If the string preceding the double bond (`=C`) contains another capital `C`, it is counted as a non-aromatic double bond (e.g., `C=C` in `CC=CCCC`).

ii. Ring systems: If the string preceding the double bond contains a numeric ring closure indicator (e.g., `C1=C` in `C1=CCCC1`), it is also recognized as part of a non-aromatic ring structure.

iii. Parentheses handling: The function is equipped to handle more complex structures, such as those involving nested parentheses (e.g., `C(...(... (...)...)...)=C` in `CC(C(C(CC)C)C)=CC`). In SMILES, nested parentheses represent branching in a molecule occurring when an atom in the main chain of the molecule is connected to one or more side chains. The parentheses indicate the start and end of each branch, and nesting occurs when a branch contains another branch. It uses the `find_opening_parenthesis` function to locate the corresponding opening parenthesis and verify that the bond belongs to a non-aromatic system.

This approach ensures that double bonds within non-aromatic systems are accurately counted, even in cases where the SMILES notation involves branching or ring structures.

### 2.1.5. Alkyl hydroxyl and hydroperoxide groups

The function `hydroxyl_group(s)` is designed to identify hydroxyl groups (–OH) present in the SMILES representation of a compound. To ensure accurate detection, the function incorporates specific conditions to avoid miscounting hydroxyl groups that are in specific arrangements that do not denote targeted simple hydroxyl groups. For example, it ensures that carboxylic acids with hydroxyl groups connected to carbonyls (C=O) are not mistakenly counted as plain hydroxyls. One important condition is that the `C(O)` pattern (a hydroxyl group in SMILES) must not be followed by `=O` or `(=O)`, which would indicate a carboxylic acid rather than a hydroxyl group. The following steps outline how this function operates to ensure accurate detection of hydroxyl groups:

- Ring number detection: The `find_cycle_number(s)` is utilized to determine the presence of cyclic structures to establish whether the hydroxyl group is part of a ring or a straight-chain structure.
- Pattern matching: The function examines the SMILES string for various patterns that denote hydroxyl groups, primarily focusing on terminal hydroxyls, where the function checks if the hydroxyl group appears at the end of the SMILES string, represented as

'O' or 'C–O' configurations (e.g., `CCCCO`) or branching with parenthesis (e.g., `CCC(C)O`). It evaluates whether the hydroxyl group appears within a branching structure or a cyclic component, recognizing patterns such as `C(...)O` or `C1(...)O`. Moreover, the function is equipped to deal with complex SMILES representations involving nested parentheses, ensuring that all possible hydroxyl configurations are evaluated (e.g., `(O)` in `CC(O)CC` and `CC(C)(O)CC`).

- Conditions for counting: Specific conditions are implemented to avoid miscounting hydroxyl groups that are either in specific arrangements that do not denote targeted hydroxyl group or are redundant in cyclic structures. In this regard, the function ensures that carboxylic acids with hydroxyl groups connected to carbonyls (C=O) are not mistakenly counted as plain hydroxyls. For example, in the SMILES representation `C(O)=O`, the hydroxyl group is not counted because it is directly bonded to a carbonyl group. As an example, one condition to take into consideration is that the `C(O)` pattern (a hydroxyl group in SMILES) does not proceed with `=O` and `(=O)`. Additional criteria influencing this classification are embedded in the code.

It's important to note that the detection of hydroperoxide groups (–OOH) in VaPOrS follows a similar pattern as that used for hydroxyl groups (–OH). The key difference in recognizing hydroperoxide groups is that instead of detecting single oxygen atoms (O), the algorithm would look for two consecutive oxygen atoms (OO). This adjustment is made throughout the function `hydroxyl_group(s)` and is provided in a new function as `hydroperoxide_group(s)` to recognize and count hydroperoxide groups in addition to hydroxyl groups.

In cyclic structures, hydroxyl groups are excluded if they are connected directly to aromatic rings (e.g., `c1ccc(O)cc1`). This is due to them being considered as another essential functional group called aromatic hydroxyl in the SIMPOL method. The sequential subsection explains how this functionality is described and detected by the developed algorithm.

*2.1.6. Aromatic hydroxyl group*

The function `aromatic_hydroxyl_group(s)` scans the SMILES string for the presence of hydroxyl groups attached to aromatic systems. The function looks for specific patterns that indicate an aromatic hydroxyl group, focusing on the position of the hydroxyl group in the SMILES string:

i. Hydroxyl at the end: The function checks if the SMILES string ends with `O`, ensuring that the preceding character(s) form part of an aromatic ring. For the SMILES string such as `c1cccc1O`, where the hydroxyl group is at the end of the SMILES, it detects the hydroxyl group attached to an aromatic ring (`c1`).

ii. Hydroxyl at the start: The function checks if the SMILES starts with the `Oc1` pattern, indicating a hydroxyl group attached to the first aromatic ring in the compound. `Oc1cc` would be recognized as having an aromatic hydroxyl group at the start.

iii. Hydroxyl in the middle: The function searches for occurrences of the pattern `c(O)` or `c(...O)` where a hydroxyl group is attached to an aromatic ring somewhere in the middle of the SMILES. For the SMILES `c1cc(O)cc1`, the hydroxyl group is identified within the ring. In strings such as `c1c(c...c1O)C`, the hydroxyl group connected to the aromatic ring `c1` is correctly identified.

iv. Hydroxyl at the end of a branch: The function checks for cases where a hydroxyl group is part of a branch but still attached to an aromatic ring. For example, in `c1c(c...)O`, where the hydroxyl group is at the end of a branch attached to an aromatic ring, it detects the correct structure.

v. Handling nested parentheses: If multiple parentheses are involved, the function ensures that the correct ring and attachment points are identified. For example, the pattern `...c1...(...c1)...)O)...` appearing in SMILES like `CC(c1cc(c(cc1)CC(C=O)C)O)C` identifies the aromatic hydroxyl correctly.

vi. Hydroxyl connected to the index of aromatic carbon: The function ensures that hydroxyl groups are also counted if they are directly attached to index of carbon atom in an aromatic ring (represented by `c` in SMILES notation). In `c1(O)cccc1`, the hydroxyl group attached to `c1` is correctly counted.

As the final filtration step, after identifying a hydroxyl group in the aromatic ring, the algorithm also checks for the presence of a nitro group within the same aromatic ring. If a nitro group is found in addition to the hydroxyl group, the compound is no longer classified as an aromatic hydroxyl group; instead, it is categorized as a nitrophenol compound. A detailed explanation of this functional group is provided in a subsequent subsection.

*2.1.7. Aldehyde group*

The function `aldehyde_group(s)` is tasked with locating aldehyde groups (i.e., terminal carbonyl groups) within a given SMILES string. The following steps illustrate how this function operates:

- Pattern matching: The function checks the SMILES string for various patterns that indicate the presence of aldehyde groups, focusing on:
    i. Aldehyde at the beginning: The function checks if the SMILES starts with common aldehyde patterns. For instance, In `O=CC`, `C(=O)C`, `O=Cc`, and `C(=O)c` patterns appearing in the SMILES, the function counts these as aldehydes.
    ii. Aldehyde at the end: The function checks if the SMILES ends with the characteristic `C=O` pattern. For example, the last three characters are checked to be `C=O`. If preceded by a `C`, such as in `CC=O`, it counts as an aldehyde. As another example with cyclic structures, for `C1C=O` as the last characters of a string where the aldehyde is linked to a cyclic structure, this is also counted.
    iii. Aldehyde in the middle of the SMILES notation (not structure): The function scans for `C=O` patterns within the SMILES string. For the structure such as `C(C=O)...`, the carbonyl group may appear in the middle of the SMILES string. However, this still corresponds to a terminal carbonyl group in the molecular structure, i.e., an aldehyde and not a ketone. This distinction is important: while the SMILES position may suggest a non-terminal group, the actual bonding context in the molecular structure confirms its identity as an aldehyde.
    iv. Aldehyde at the end of a branch: The function examines occurrences of `C=O)` at the end of branches. For example, the appearance of `CC=O)...` pattern in the SMILES counts it as an aldehyde.
- Conditions for counting: Specific conditions are established to ensure accurate counting and avoid misidentification of aldehyde groups:
    i. Connected to carbon atoms: The function ensures that aldehydes are counted only if they are connected to carbon atoms directly. In `C(C=O)...` pattern, for example, the `C` preceding the `C=O` indicates a valid aldehyde.
    ii. Handling cyclic structures: The function accounts for rings, ensuring that aldehydes connected to cyclic structures are counted appropriately. For example, in `C1(...)C=O`, the presence of `C1` before the aldehyde indicates a connection to a ring, thus counting it as an aldehyde.

iii. Parentheses handling: The function effectively manages nested structures, checking for relevant connections before counting. As an instance, if `C=O` is branched from a carbon e.g., in `C(...)(C=O)...` pattern, it counts as an aldehyde.

### 2.1.8. Carboxylic and peroxy acid groups

The function `carboxylic_acid_group(s)` is designed to identify the presence of carboxylic acid groups within a compound. The detection is briefly broken down into the following steps:

- Carboxylic acid at the beginning: The function begins by checking if the SMILES string starts with the characteristic patterns for a carboxylic acid group. Allowed prefixes are `O=C(O)` or `OC(=O)`. For example, in the SMILES string `O=C(O)CC`, the carboxyl group `O=C(O)` at the beginning of the molecule is detected.
- Carboxylic acid as the last characters: The function checks if the SMILES string ends with patterns such as `C(=O)O` and `C(O)=O`, indicating a carboxylic acid group at the end of the molecule. As an instance, for the compound `CCC(=O)O`, the carboxylic acid group at the end is correctly identified as `C(=O)O`.
- Carboxylic acid in the middle of the SMILES: The function searches for occurrences of the carboxylic acid group in the middle of the SMILES string using the patterns like `C(=O)O` and `C(O)=O`. Each time one of these patterns is found, the function increases the count of carboxylic acid groups. For instance, in the SMILES `CC(C(=O)O)C(C(O)=O)C`, the carboxylic acid groups `C(=O)O` and `C(O)=O` in the middle are detected.

The detection of peroxy acid groups follows a similar pattern to that used for carboxylic acid groups within the model. The key difference in recognizing peroxy acid groups lies in the algorithm's adjustment to look for two consecutive oxygen atoms e.g., `C(=O)OO` and `C(OO)=O` instead of single oxygen atoms found in carboxylic acids e.g., `C(=O)O` and `C(O)=O`. By implementing this modification throughout the model, peroxy acid functional groups can be counted as well.

### 2.1.9. Ketone group

The function `ketone_group(s)` is designed to identify and count the presence of ketone groups in the SMILES string of a compound. These patterns can be located at the start, middle, or end of the SMILES string, as well as within rings. The detection process is divided into several steps:

- Ketone as the first characters: The function checks if the SMILES string begins with a ketone pattern (e.g., `O=C(C...)C` in `O=C(CCC)C` or `O=C(c...)C...` in `O=C(c1ccccc1)CCO`). These patterns represent the ketone group at the beginning of the SMILES string, followed by either a non-aromatic or aromatic carbon. The function also handles ketones connected to rings, such as `O=C1...` in `O=C1CCCC1` indicating a carbonyl group attached to the first position of a ring. Note that although the ketone group appears at the start of the SMILES string, it may not be at the beginning of the molecular structure itself. The pattern is recognized based on bonding context, not string position.

- Ketone as the last characters: The function checks if the SMILES string ends with `=O`, indicating a ketone group at the end of the molecule. If the ketone is ring-connected (e.g., `C1CCCC1=O`), additional checks ensure the presence of a carbonyl group within the ring. Similarly, the ketone group may appear at the end of the SMILES notation, but in the molecular structure, it could be part of a cyclic or internal configuration. The detection logic is based on chemical connectivity, not linear SMILES order.

- Ketone in the middle of the SMILES: The function searches for ketone groups within the middle of the SMILES string using patterns such as `C(=O)`, representing a carbonyl group between two carbons. Also, the function carefully checks if the ketone is branched, ensuring accurate identification of the ketone group in the middle. As an example, in the SMILES string `CC(=O)C`, the middle ketone group `C(=O)` is detected between two carbon atoms.

- Handling parentheses: The function accounts for complex SMILES structures that contain nested parentheses. It ensures that ketone groups within branches (e.g., `...C(C(C...)=O)...` in `CC(C(CCC)=O)CC`) are properly identified by finding the matching opening and closing parentheses.

*2.1.10. Open-chain, alicyclic, and aromatic ether groups*

Three functions `open_chain_ether(s)`, `alicyclic_ether(s)`, and `aromatic_ether(s)` are defined to distinguish between ethers present in the SMILES notation (open-chain, alicyclic, and aromatic ethers). Below is a breakdown of the major components and logical flow within this algorithm.

A key part of the function's operation involves detecting in-ring ethers. The function starts by searching for the sequence "`OC`" within the SMILES string, which is the primary sequence of the ether functional group. The other carbon atom bonded to the oxygen atom of the "`OC`" sequence (e.g., `COC` or `C(OC…)`) is then searched to identify if the ether function is alicyclic or open-chain. For each occurrence, the algorithm checks if the sequence is part of a non-aromatic ring by comparing its position relative to the numerical ring indicators. This is done by locating the positions of ring closure numbers (e.g., `C1...C1`) relative to the ether group. If the ether group lies outside the ring closure points (e.g., `C1CCCC1COCC`), it is classified as an open-chain ether. On the other hand, if the ether group is found within the two identical ring numbers (e.g., `C1...COC..C1`), the algorithm hesitates if the ether is part of the ring or an open-chain type. Therefore, the function employs a nested structure-parsing approach to clarify this issue. It uses parentheses to detect branching points or nested structures within the molecule. The algorithm carefully traces the boundaries of rings and other nested structures simultaneously. If the ether group is embedded within parentheses and surrounded by ring numbers (e.g., pattern `C1...(...COC...)...C1` in SMILES `C1CC(COC)CC1`), it is open-chain ether, while if it follows patterns, such as `...C1...(...COC...C1...)...`, it is alicyclic (e.g., SMILES `C1C(COCC1C)CC`). On the other hand, the function also checks if the ether is located near the start or end of a ring closure, which would indicate that the ether is alicyclic (e.g., pattern `C1...C(...)O1`), otherwise, it would be an open-chain ether. In compounds containing multiple rings, the function iterates through each ring. It systematically searches for ether groups within and around each ring, ensuring that all possible locations are checked for ether group presence. If the ether group is attached to a non-aromatic ring, special handling is performed to ensure accurate detection.

On the other hand, for aromatic ethers, the model looks for occurrences of `'Oc'` within the SMILES string. The logic checks different cases based on what precedes `'Oc'`:

- Simple alkyl group (`'C'` before `'Oc'`): This detects linear aromatic ethers where the oxygen is connected directly to an alkyl group (e.g., `COc1ccccc1`).

- Nested structures with parentheses: The model handles cases where the ether group is part of a complex structure. If the ether group is surrounded by parentheses, the model traces the original alkyl chain, ensuring it connects to an aromatic ring (e.g., patterns `C(Oc` and `C(...)Oc` in SMILES `CC(Oc1cccc1)CCC` and `CC(C(O)=O)Oc1cccc1`).

- In case the ether group is attached to one aromatic and one non-aromatic ring, the model iterates over possible ring numbers to find ether groups (e.g., `'C1CCCC1Oc1ccccc1'` where `'C1'` is part of a ring and `'Oc1'` is the ether group). The same logic is applied to nested structures with parentheses, ensuring that the correct connection between alkyl and aromatic groups is maintained.

- The model repeats the search, but this time looking for `'OC'`, where the aromatic group comes first (e.g., `'c1ccccc1OC'` where `'c1'` indicates an aromatic ring and `'OC'` is the ether). Similar checks are performed for direct aromatic ether group connection (`'c1OC'`), complex nested structures (e.g., `c1(OC)ccc1` and `c1cc(OC)cc1`), and parentheses-based structures (e.g., `c1cc(ccc1)OC`).

*2.1.11. Ester and nitroester groups*

The developed function, `ester_group(s)`, identifies and quantifies ester groups within a given SMILES string. The ester group is characterized by the bonding of a carbonyl group (C=O) to an alkoxy group (O–R). This pattern is typically (not always) represented in SMILES as `C(=O)O`, and variations in its placement within the molecule must be accounted for. For instance, when analyzing the ester group, if the alkoxy group (-OR) shows up in the SMILES first and then the carbonyl group (C=O), common patterns, such as `...C(...)(...)OC(=O)C...` (e.g., in `CC(CO)(C)OC(=O)CC` with bolded characters) and `...C(...)(OC(=O)C...)...` (e.g., in `CC(CO)(OC(=O)CC)C` with bolded characters) are identified. Conversely, if the SMILES notation proceeds from the acid-side carbon (i.e., carbon attached directly to the carbonyl group in the ester bond –C(O)OR), ester groups may be represented as `...C(...)(...)C(=O)OC...` (e.g., in `CC(CO)(C)C(=O)OCC` with bolded characters) or `...C(...)(C(=O)OC...)...` (e.g., in `CC(CO)(C(=O)OCC)C` with bolded characters). These patterns ensure that ester groups are recognized irrespective of their position within a molecule.

The above-mentioned patterns with several other ones related to ester group may be determined whether at the start, middle, or end of the SMILES string:

- If the ester group is at the beginning, the function looks for patterns such as `O=C(C...)OC...` and `O=C(OC...)C` (e.g., in SMILES `O=C(CCO)OCCC` and `O=C(OCCC)CCO`), where the group starts with the double-bond oxygen atom.

- For esters embedded within the middle of a SMILES string, the function searches for the ester signatures such as `C(=O)O` and `OC(=O)`, confirming that the carbonyl group is bonded to the oxygen atom of the alkoxy group and followed by a suitable molecular fragment. For instance, in a SMILES notation like `CCC(=O)OCC` and `CCOC(=O)CC`, the ester group is correctly identified as part of the main chain.

- If an ester group is located at the end of the SMILES string, the function identifies patterns such as `)=O` sequence as an indicator of the terminal carbonyl group, followed by the appropriate bonding structure. This ensures that molecules like `CC(OCC)=O` are correctly parsed with the ester group assigned to the end of the chain.

- In cases of aromatic esters, where alternating single and double bonds are common, the function adapts the detection logic to properly handle aromaticity. For example, it correctly identifies the ester group in a molecule like `O=C(OCC)c1ccccc1`, where the ester is attached to an aromatic benzene ring.

- The algorithm is designed to detect esters even in highly branched molecules. For instance, in a molecule like `CCC(C(=O)OCCC)(C)CC`, the ester group is part of a branching structure, and the function ensures it is accurately parsed by considering the nested arrangement of atoms.

On the other hand, the model locates the starting and ending positions of the acid-side branch for the ester group and searches for the presence of a nitro group (N(=O)=O) in the branch. If a nitro group is detected, the compound is classified as a nitroester, and the `nitroester_number` is incremented. In cases where no nitro group is present, the compound is classified as a regular ester, and the `ester_number` is incremented accordingly.

*2.1.12. C=CC=O in non-aromatic rings*

The function `nonaromatic_CCCO(s)` is designed to quantify C=C–C=O substructures in a molecular SMILES notation. This section explains how the function works.

Since the substructure of interest occurs within rings, the function first checks whether the molecule contains any cyclic structures. The function uses `find_cycle_number(s)` to determine if any rings exist in the molecular structure. If no rings are found, the function

terminates early. When rings are detected, the function proceeds to locate their positions within the SMILES string.

The next step is to detect the C=C–C=O group within the identified rings. This involves scanning the part of the SMILES string that represents each ring and checking for the specific pattern of atoms and bonds. Several important aspects are considered during this analysis:

- Pattern search: Within the extracted portion, the function looks for patterns that match the C=C–C=O group. For example, the presence of `C=O` at the start of a ring followed by a conjugated double bond (`C=C`) within the ring (e.g., pattern `O=C1...C(...)=C1` bolded in SMILES `O=C1CCC(OC)=C1C`), or variations where the C=C and C=O groups may be spaced by additional atoms, or where they may appear in different locations within the ring (e.g., pattern `...1...C(=O)C(...)=C...1...` bolded in SMILES `O=C1CC(=O)C(CC)=CC1C`).
- Handling complex ring structures and nested rings: The function accounts for these complexities by carefully navigating through the parentheses and ensuring that the entire cyclic structure is examined for the C=C–C=O group (e.g., pattern `...C(=C(C(...1...)=O)...)...` bolded in SMILES `CC1C(=C(C(CC1C)=O)C)CC`).

### 2.1.13. Nitrate group

The `nitrate_number(s)` function identifies and quantifies nitrate functional groups within the provided SMILES notation. This function examines specific configurations characteristic of nitrate, including the standard nitrate structure `ON(=O)=O`, the quaternary form `O[N+](=O)[O-]`, the N-nitro structure `O=N(=O)O`, and variations like `O=[N+]([O-])O` and `[O-][N+](=O)O`, which collectively reflect diverse bonding scenarios in nitrate chemistry. The function calculates the count of each of these configurations and aggregates them to derive the total number of nitrate groups present.

### 2.1.14. Nitro group

The identification of nitro groups is executed through the `nitro_group(s)` function, which analyzes the SMILES notation to quantify nitro functional groups. This function employs a series of search operations to locate specific nitro structures, notably `N(=O)=O`, `O=N(=O)`, and various ionic forms such as `[N+](=O)[O-]`, `O=[N+][O-]`, and `[O-][N+](=O)`. Each search utilizes a loop that not only finds occurrences of these structures but also ensures that adjacent

atoms do not disrupt the nitro configuration—specifically, it checks that there is no oxygen atom directly connected to the nitro group, which would suggest an alternative bonding scenario (i.e., nitrate group). As the final filtration step, the algorithm verifies whether the identified nitro group is not part of an aromatic ring (i.e., benzene). If this condition is met, it then checks for the presence of a hydroxyl group in the ring according to section 2.6. If no hydroxyl group is found, the nitro group is counted independently. However, if a hydroxyl group is present, the compound is categorized as a nitrophenol group. A detailed explanation of this functional group is provided in a subsequent subsection.

### 2.1.15. Nitrophenol

The function `nitrophenol_group(s)` is designed to identify and count the number of nitrophenol groups. After identifying aromatic hydroxyl group, the function then checks for the presence of a nitro group in the same aromatic ring within the SMILES string:

- As the last character sequence: It inspects whether the string ends with the motif `N(=O)=O`, which signifies a nitrophenol. The function checks if it is connected to the aromatic cyclic structure by examining the characters before it. For example, if the character is an aromatic ring identifier, the counter is incremented accordingly (e.g., `c1cc(O)ccc1N(=O)=O`).
- As the first character sequence: The function checks if the string starts with the pattern `O=N(=O)c1`, indicating a nitrophenol positioned within the SMILES string (e.g., `O=N(=O)c1cc(O)ccc1`). If this pattern is detected, the counter is incremented.
- As middle character sequences: A loop is employed to search for occurrences of patterns such as `c(N(=O)=O)`, which indicates that nitrophenol is positioned within the structure (e.g., `c1cc(N(=O)=O)cOcc1O`). Each time this pattern is found, the counter is increased. The function also iterates through the previously established list of rings to search for patterns of the form `j(N(=O)=O)` (where `j` is an aromatic carbon identifier). Whenever a match is found, the counter is incremented (e.g., `c1(N(=O)=O)cc(O)ccc1`).

### 2.1.16. Acylperoxynitrate group

The function `carbonylperoxynitrate_group(s)` is designed to detect the presence of acylperoxynitrate groups within a given SMILES string `s`. This function systematically counts occurrences of several distinct structural motifs characteristic of acylperoxynitrates, including

`OON(=O)=O`, `OO[N+](=O)[O-]`, `O=N(=O)OO`, `O=[N+]([O-])OO`, and `[O-][N+](=O)OO`. The function aggregates the counts of these motifs into a single variable, `carbonylperoxynitrate`, which represents the total number of acylperoxynitrate groups identified.

*2.1.17. Peroxide group*

The function `peroxide_group(s)` is engineered to identify and quantify peroxide groups. The function first determines the number of cyclic structures within the SMILES string by calling the `find_cycle_number(s)` function. If cyclic structures are found, a list of ring identifiers, comprising both carbon (`C`) and aromatic (`c`) rings, is generated to facilitate later checks. Subsequently, the function employs a loop to search for occurrences of the `OOC` motif, which signifies the presence of peroxy groups. Each iteration of the loop calls `s.find('OOC', ...)` to locate the next occurrence of the motif. Multiple conditions are assessed to ensure that the identified `OOC` is correctly positioned relative to other atoms or rings:

- Adjacent carbon or aromatic carbon atoms: If the character preceding `OOC` is `C`, `Cx` or `cx` (`x=1,2,…` is cyclic index), indicating that `OOC` is bonded to a carbon atom, the `peroxide_number` is incremented (e.g., `CCCOOCC`, `C1CCCC1OOCC` and `Cc1cccc1OOC`).
- Branching structures: If the character before `OOC` is a parenthesis, the function retrieves the index of the last corresponding parenthesis before `OOC` and verifies that the atom preceding this parenthesis is a carbon atom or an aromatic ring. If so, the counter is increased. For example, when the character before `OOC` is a closing parenthesis, the function checks whether the `OOC` is connected to a carbon atom in a similar manner as described previously. This involves searching back to the last opening parenthesis and ensuring the atom connected to that parenthesis is a carbon atom or part of a cyclic structure (e.g., pattern `...C(...)OOC...` bolded in SMILES `CCC(CO)OOCC`).

These checks comprehensively ensure that only valid peroxide groups are counted, accounting for the complex connectivity possible within SMILES representations. The function ultimately returns the `peroxy_number`, providing a quantitative measure of peroxide groups within the molecular structure.

*2.1.18. Aromatic amine group*

Following the identification of aromatic rings, the function `aromatic_amine_group(s)` locates nitrogen atoms (`N`) within the SMILES string and determines their bonding to aromatic carbons.

- Direct bond to aromatic carbon (`cN`): If nitrogen (`N`) is found immediately following an aromatic carbon (`c`), it is counted as part of an aromatic amine group. To ensure the nitrogen atom belongs to an aromatic amine and not a nitro group (-NO2), the model incorporates an additional condition.
- Nitrogen in numbered rings (`c1N`, `c2N`, etc.): If nitrogen is attached to a carbon in a numbered ring, such as `c1N`, the nitrogen is identified as part of an aromatic amine group.
- Parenthetical structures (`c(...)N`): In cases where nitrogen is attached within parentheses following a cyclic group, the function checks if the nitrogen is part of an aromatic ring by verifying the bonding pattern of the cyclic carbon to the nitrogen. Parentheses in SMILES represent branching, and this function ensures that any branching nitrogen groups attached to aromatic carbons are also detected.

The function iterates through the SMILES string to ensure that all occurrences of nitrogen atoms are evaluated. The bonding pattern of each nitrogen atom is assessed against the aromatic rings identified in the first step. If the nitrogen atom is confirmed to be attached to an aromatic ring and not part of a nitro group, it is counted as part of an aromatic amine group. The total count of such groups is stored in the variable `aromatic_amine_number` and returned as the output of the function.

*2.1.19. Primary, secondary, and tertiary amide and amine groups*

The model is designed to detect and count primary, secondary, and tertiary amide and amine groups in a molecule represented by a SMILES string. A primary amide has the functional group structure –C(=O)NH2, and the model identifies both simple and branched forms of this group. To differentiate between primary amides and primary amines, the model specifically excludes patterns without a double-bonded oxygen, i.e., –C(...)NH2 where '...' is not =O, thereby ensuring correct identification of amide groups versus amine counterparts.

- The function first identifies primary amide groups located at the beginning of the SMILES string by searching for patterns such as `O=C(N)` or `NC(=O)`, as seen in SMILES

O=C**(N)**CCCO and **NC(=O)**CCCO). It also detects primary amides at the end of the SMILES string by tracing patterns like C(=O)N and C(N)=O, as in CCOC**C(=O)N** and CCOC**C(N)=O**. To identify primary amine groups at the beginning or end of the strings, the function checks for similar patterns but without a carbonyl group (=O). In other words, if a nitrogen is bonded to a carbon that does not carry a =O, it is interpreted as an amine rather than an amide. For example, **C(N)**CCCO, **NC**CCCO, CCOC**CN**, and CCOC**CN** are recognized as containing primary amine groups.

- The model also identifies primary amides within branches or internal positions in the molecule by searching for specific patterns, such as C(=O)N  and C(N)=O, as seen in the SMILES strings  CC(**C(=O)N**)CC  and CC(**C(N)=O**)CC. To identify primary amine groups in similar positions, the model checks for the absence of the double-bonded oxygen (=O) on the carbon adjacent to the nitrogen. This results in patterns like CN) in CC(**CN)**CC  and CC(**CN)**CC.

For secondary and tertiary amides, the model similarly searches for patterns where the nitrogen atom is bonded to two or three carbon atoms, respectively. Secondary amides have the structure –C(=O)NR, where R represents an alkyl group starting with a carbon atom attached to the nitrogen atom, and tertiary amides have the structure –C(=O)N(R)R', where both R and R' are alkyl groups starting with carbon atoms attached to the nitrogen atom. The model identifies these structures by recognizing the presence of additional carbon attachments to the nitrogen atom. For secondary amides, the function searches for patterns that indicate the nitrogen is bonded to one additional carbon group, distinguishing them from primary amides by checking for two single bonds to nitrogen, along with the carbonyl group. Similarly, for tertiary amides, the function detects two alkyl groups attached to the nitrogen atom in addition to the carbonyl group. Once these amide patterns are identified, the model applies the same exclusion method for the double-bonded oxygen, converting these amides into their corresponding secondary and tertiary amines. For secondary amines, the nitrogen is attached to two carbon atoms, and for tertiary amines, the nitrogen is bonded to three carbon atoms. This method ensures that the correct amide or amine group is identified and classified, whether it is primary, secondary, or tertiary, based on the number of carbon attachments to the nitrogen atom.

Finally, the model counts the number of carbon atoms in the secondary and tertiary amides that are not part of the R and R' groups in the structure. This count is considered as the number of carbons on the acid side of the amides. For primary amides, since there are no additional alkyl

groups attached to the nitrogen atom, all carbon atoms in the structure are considered to be on the acid side of the amide. This ensures accurate categorization and counting of carbon atoms associated with the amide's acid side, contributing to the overall structural analysis of the molecule.

### 2.2. Saturation vapor pressure calculation

Once functional groups are identified, VaPOrS converts the detection results into integer values representing the frequency of each group in a given compound. These values are stored in an array and serve as the critical input for property prediction. In the present implementation, saturation vapor pressure is calculated using the SIMPOL group-contribution method, in which the total logarithmic vapor pressure is expressed as the sum of contributions from all relevant functional groups as a function of temperature. In the model:

i.   Matrix B is read from a pre-defined text file containing coefficients of $B_{k,1}$, $B_{k,2}$, $B_{k,3}$, and $B_{k,4}$ for each functional group k, according to Table 5 of (Pankow and Asher 2008). A select functional group is assigned a value for contribution to saturation vapor pressure in the $i^{th}$ SMILES string (such as hydroxyl, aldehyde, and ketone groups, etc.).

ii.  Then, $b_k(T)$ and $P_{L,i}^0(T)$ are calculated for any given temperature according to Equations 1 and 2. The total liquid (saturation) vapor pressure, $P_{L,i}^0$ (atm), is calculated as the sum of all functional group contributions.

$$b_k(T) = \frac{B_{k,1}}{T} + B_{k,2} + B_{k,3}\, T + B_{k,4}\, \ln T \tag{1}$$

$$\log_{10} P_{L,i}^0(T) = \sum_k \nu_{k,i}\, b_k(T) \quad k = 0, 1, 2, \ldots \tag{2}$$

where $\nu_{k,i}$ is the number of groups of type k, $b_k(T)$ is the contribution to $\log_{10} P_{L,i}^0(T)$ by each group of type k, and $T$ is the temperature. Also, $0$ and $L$ show the reference and Liquid.

iii. To fit the saturation vapor pressure data to the Antoine equation (Equation 3) to enable further use of the saturation vapor pressure values in different applications, the saturation vapor pressure is calculated at 1000 temperature points across a wide range

of temperatures from 220 K to 450 K according to Equation 2. The obtained data are

then used in a non-linear least squares fitting procedure, which minimizes the difference

between the data and the saturation vapor pressure values predicted by the Antoine

equation. The Antoine equation parameters (i.e., A, B, and C) are then obtained for each

compound.

$$\log P_{sat} = A - B/(T + C) \tag{3}$$

iv.    After obtaining the Antoine equation parameters, the vaporization enthalpy relationship

can be derived using the Clausius-Clapeyron equation:

$$\frac{d \log P_{sat}(T)}{d(\frac{1}{T})} = -\frac{\Delta H_{vap}(T)}{2.303R} \tag{4}$$

$$\Delta H_{vap}(T) = -2.303R\left(\frac{d \log P_{sat}(T)}{d(\frac{1}{T})}\right) \tag{5}$$

Here, $\Delta H_{vap}(T)$ is the temperature-dependent enthalpy of vaporization, and $R$ is the universal

gas constant. This expression relates the slope of the logarithm of the saturation pressure with

respect to the inverse of temperature to the enthalpy of vaporization. The Antoine equation

provides a framework to calculate saturation vapor pressure at any given temperature, and this

relationship extends the utility of the model by allowing the determination of thermodynamic

quantities such as vaporization enthalpy.

v.    The output is written to a CSV file named by the user (e.g., output.csv), where each line

corresponds to the $i^{th}$ compound and its associated data, including its SMILES string,

the count of each functional group in its structure, its saturation vapor pressure at 300

17         K, and its fitted Antoine equation parameters.

Figure 3 illustrates a flowchart of the VaPOrS from input to output.

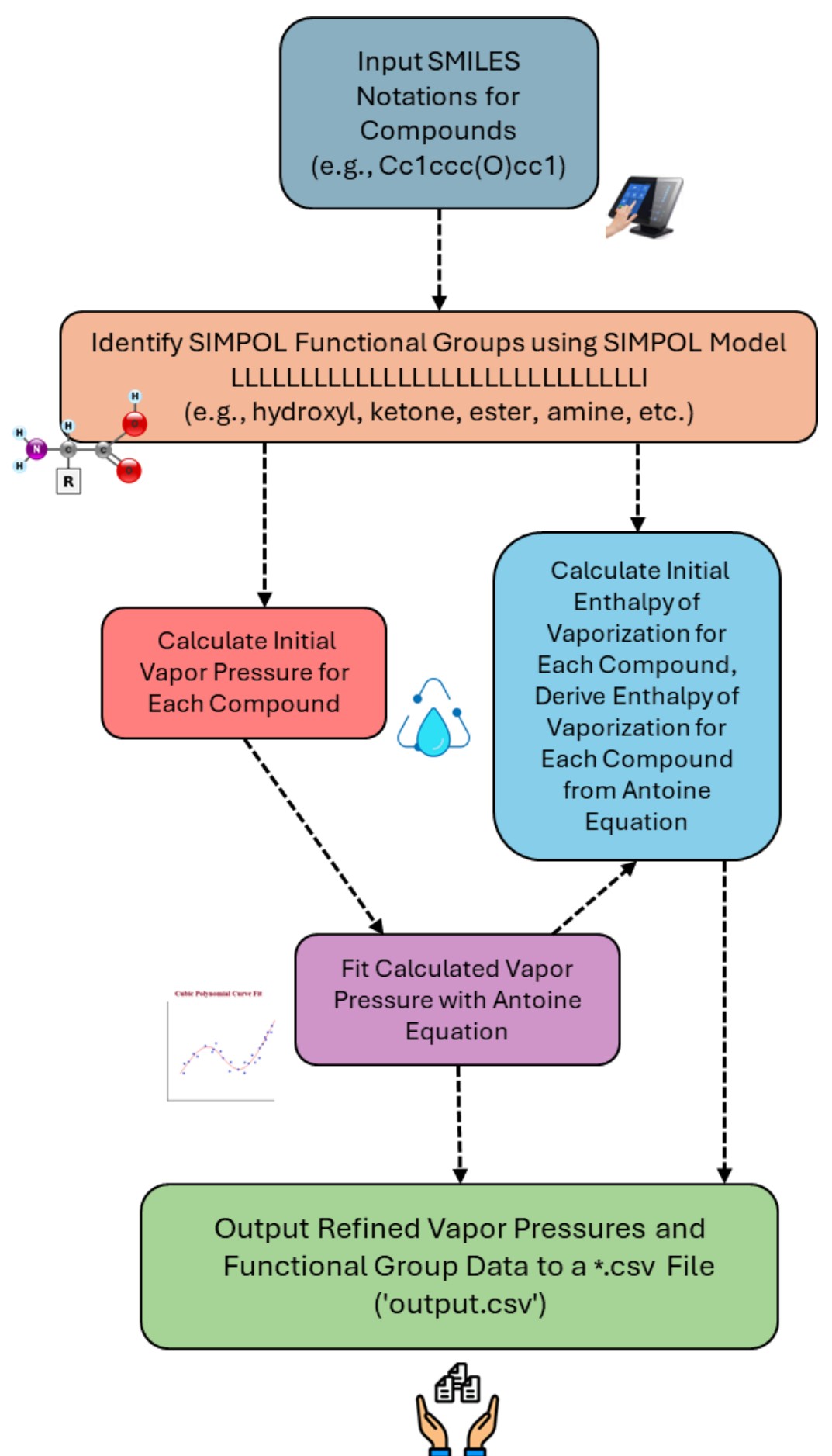

Input SMILES Notations for Compounds (e.g., Cc1ccc(O)cc1)

Identify SIMPOL Functional Groups using SIMPOL Model LLLLLLLLLLLLLLLLLLLLLLLLLLLLLLI (e.g., hydroxyl, ketone, ester, amine, etc.)

Calculate Initial Vapor Pressure for Each Compound

Calculate Initial Enthalpy of Vaporization for Each Compound, Derive Enthalpy of Vaporization for Each Compound from Antoine Equation

Fit Calculated Vapor Pressure with Antoine Equation

Output Refined Vapor Pressures and Functional Group Data to a *.csv File ('output.csv')

**Figure 3.** Flowchart of the automated process of VaPOrS, beginning with SMILES notation input, followed by the identification of functional groups, initial saturation vapor pressure and enthalpy of vaporization calculations, their fitting to the Antoine equation, and final output.

## 3. Validation

### 3.1. Saturation vapor pressure

To evaluate the accuracy of the developed automated VaPOrS in predicting the saturation vapor pressures of organic compounds, the model was applied to a subset of the original dataset used to develop the SIMPOL parameterization. Saturation vapor pressures for 224 organic compounds were calculated using VaPOrS and compared against experimental data to assess the accuracy of the implementation and ensure consistency with established results.

Figure 4 presents a comparative analysis between the saturation vapor pressures computed by the VaPOrS and those reported by the SIMPOL method and measurement at a specific temperature (i.e., 333.15 K). The x-axis represents the experimental saturation vapor pressures, while the y-axis represents the numerical values calculated by the VaPOrS (filled symbols with no edge color) and the SIMPOL method (black-edged hollow symbols). Due to a complete overlap in the data points, the black-edged symbols from the SIMPOL method can be seen over the filled symbols from VaPOrS. A diagonal line is shown in the figure, indicating the ideal correlation where the computed values would perfectly match the measured saturation vapor pressures. Data points positioned close to this diagonal demonstrate a high level of agreement between the two approaches. Quantitatively, the comparison yields an RMSE of 0.4232 and an $R^2$ of 0.9648 for log P, confirming the excellent predictive performance of VaPOrS in reproducing experimental vapor pressures at this temperature.

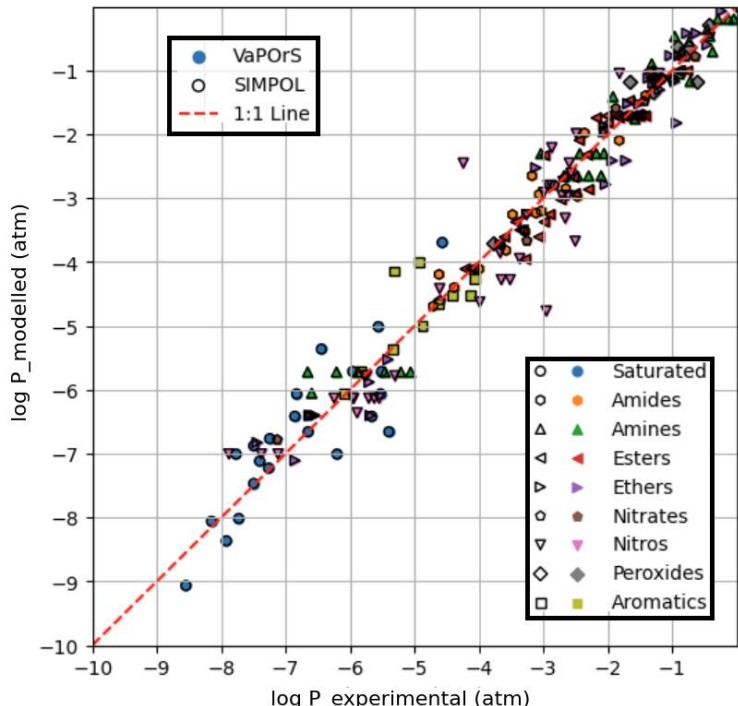

**Figure 4**. Comparison of saturation vapor pressure values calculated by the VaPOrS (filled symbols with no edge color) and the SIMPOL method (symbols with no face color and black edges) against measured saturation vapor pressures at 333.15 K. The diagonal line represents the ideal 1:1 correlation, and the proximity of data points to this line indicates the accuracy of both methods in predicting saturation vapor pressures. The overlap in symbols is visible due to the black-edged SIMPOL markers covering the filled VaPOrS symbols.

Figure 5 illustrates the saturation vapor pressure results obtained using the VaPOrS model, the SIMPOL method, and experimental measurements at seven different temperatures: 273.15 K, 293.15 K, 313.15 K, 333.15 K, 353.15 K, 373.15 K and 393.15 K. Similar to Figure 4, the x-axis represents the measured saturation vapor pressures, while the y-axis displays the values calculated by VaPOrS (filled symbols with no edge color) and the SIMPOL method (black-edged hollow symbols). This figure includes a larger dataset, allowing for a more comprehensive assessment of model performance across a range of temperatures. Many data points are clustered close to the 1:1 line, yielding an RMSE of 0.5556 and an $R^2$ of 0.9570 for log P, further confirming the effectiveness of the VaPOrS model in predicting saturation vapor pressures for various compounds across different temperatures as well. Some deviations from the line are observed e.g., the experimental values for the nitro and saturated compound saturation vapor pressures and *T*-dependencies have large uncertainties as seen in Figures 4 and 5. These discrepancies, which were present in the original dataset, do not undermine the

overall trend, which demonstrates strong agreement among the three methods and suggests that the VaPOrS code is reliable and robust across a wider temperature range. Most importantly, we see complete agreement in the output of VaPOrS and SIMPOL.

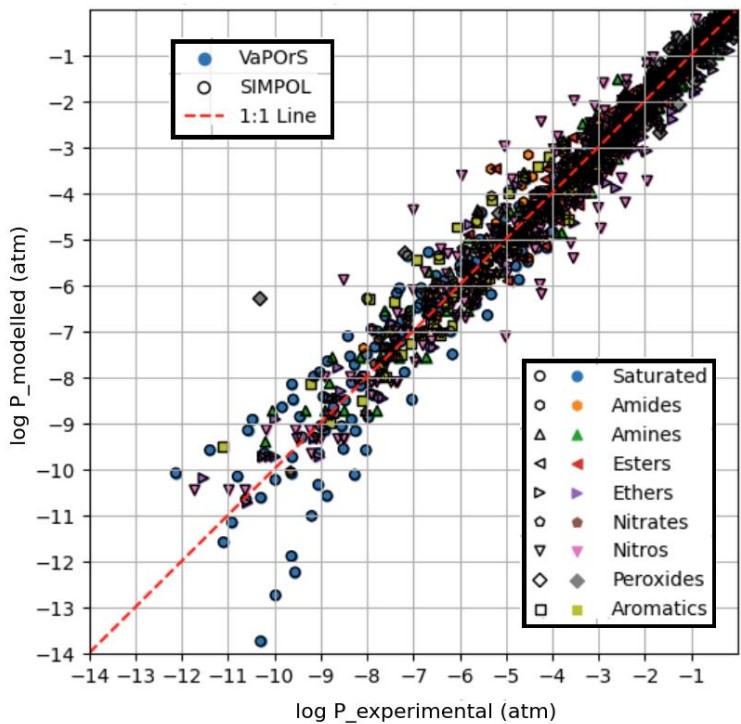

**Figure 5**. Comparison of saturation vapor pressures obtained from the VaPOrS model, the SIMPOL method, and experimental measurements at seven different temperatures (273.15 K, 293.15 K, 313.15 K, 333.15 K, 353.15 K, 373.15 K, and 393.15 K). The x-axis represents the measured saturation vapor pressures, while the y-axis shows the values calculated by VaPOrS (filled symbols) and the SIMPOL method (hollow symbols with black edges). The diagonal line indicates the ideal correlation, with points near the line demonstrating good agreement between the methods.

### 3.2. Enthalpy of vaporization

Figure 6 illustrates the relationship between the experimental vaporization enthalpy values and those calculated by both the VaPOrS and the SIMPOL method at 333.15 K. The x-axis represents the experimentally measured values, while the y-axis displays the calculated values. The results from the VaPOrS are represented by filled symbols without edge color, indicating a direct prediction from the present model. In contrast, the SIMPOL method results are depicted

as symbols with no face color and distinct black edges. The presence of overlapping symbols highlights instances where the calculated values from SIMPOL cover those from VaPOrS. Many points cluster close to the 1:1 line, but deviations are also observed. Quantitatively, the comparison yields an RMSE of 14.4740 and an $R^2$ of 0.6146 for $\Delta H$. These results indicate that while VaPOrS captures the general trend of enthalpy data, the agreement is weaker than for vapor pressures. These discrepancies could be attributed to the structural complexity of certain compounds making intramolecular interactions important and not amenable to simple group additivity predictions, which is also a known limitation in the SIMPOL method itself, or potentially, they could point out issues in the original experimental measurements.

Although the SIMPOL model does not explicitly define an applicability domain, the structure-based framework of VaPOrS offers an opportunity to explore this aspect. Since the code identifies and counts functional groups for each molecule, it can be used to highlight cases where predictions may be less reliable. For instance, compounds such as Decanedioic acid, which contains two carboxyl groups, or Hexanamide, where strong hydrogen-bonding interactions may play a role, or Diethyl-peroxide, which contains a relatively uncommon peroxide group show noticeable deviations from experimental values (See Figure 7). These examples suggest that compounds with an unusually high number of hydroxyl or carboxyl groups, or those containing less common fragments such as peroxides, often exhibit larger deviations from experimental vapor pressures. Likewise, the co-occurrence of multiple reactive groups within the same molecule (e.g., carbonyl–peroxide combinations) may introduce additional uncertainties. While a systematic applicability domain analysis is beyond the scope of this study, we note that VaPOrS could be extended to provide such functionality, thereby guiding users in assessing the reliability of SIMPOL predictions for structurally complex molecules.

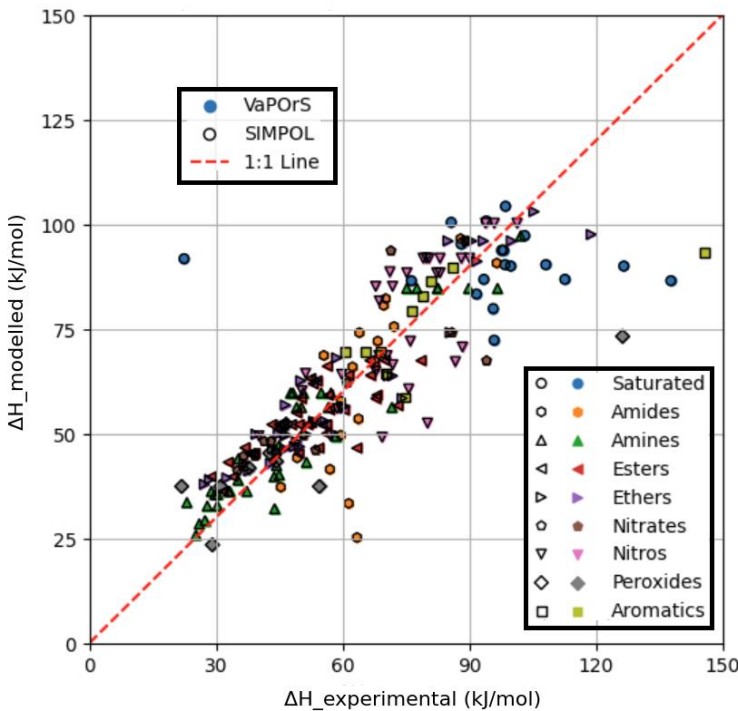

**Figure 6.** Comparison of enthalpy of vaporization values calculated by the VaPOrS (filled symbols without edge color) and the SIMPOL method (symbols with no face color and black edges) against experimental measurements at 333.15 K. The diagonal line indicates the ideal 1:1 correlation, showcasing the accuracy of the models. Overlapping symbols are observed, with black-edged SIMPOL markers obscuring the filled VaPOrS symbols.

### 3.3. Antoine equation

The results of the fitting process for nine compounds, i.e., 2-methoxy-tetrahydro-pyran, decanedioic acid, methyl-benzoate, phenylamine, hexanamide, phenylmethyl-nitrate, 2-methyl-6-nitrobenzoic acid, diethyl-peroxide, and 2-napthol as representatives of ethers, saturated, esters, amines, amides, nitrates, nitro-compounds, peroxides, and aromatics, respectively, are visualized in Figure 7. The figure illustrates the temperature-dependent behavior of both pressure and enthalpy of vaporization for Antoine and SIMPOL relationships generated by VaPOrS and compares them with experimental data across varying temperatures. The left y-axis represents the logarithmic saturation vapor pressure in atmospheres, while the right y-axis shows the enthalpy of vaporization in kJ/mol. This dual-axis representation enables a direct visual comparison between pressure and enthalpy trends as the temperature increases. The fitting results demonstrate a high degree of agreement between the Antoine and SIMPOL curves for all compound classes, implying that the Antoine equation given by VaPOrS can be applied effectively to estimate saturation vapor pressures with good accuracy across a broad

1 range of temperatures, enhancing the utility of the saturation vapor pressure data for various

2 applications.

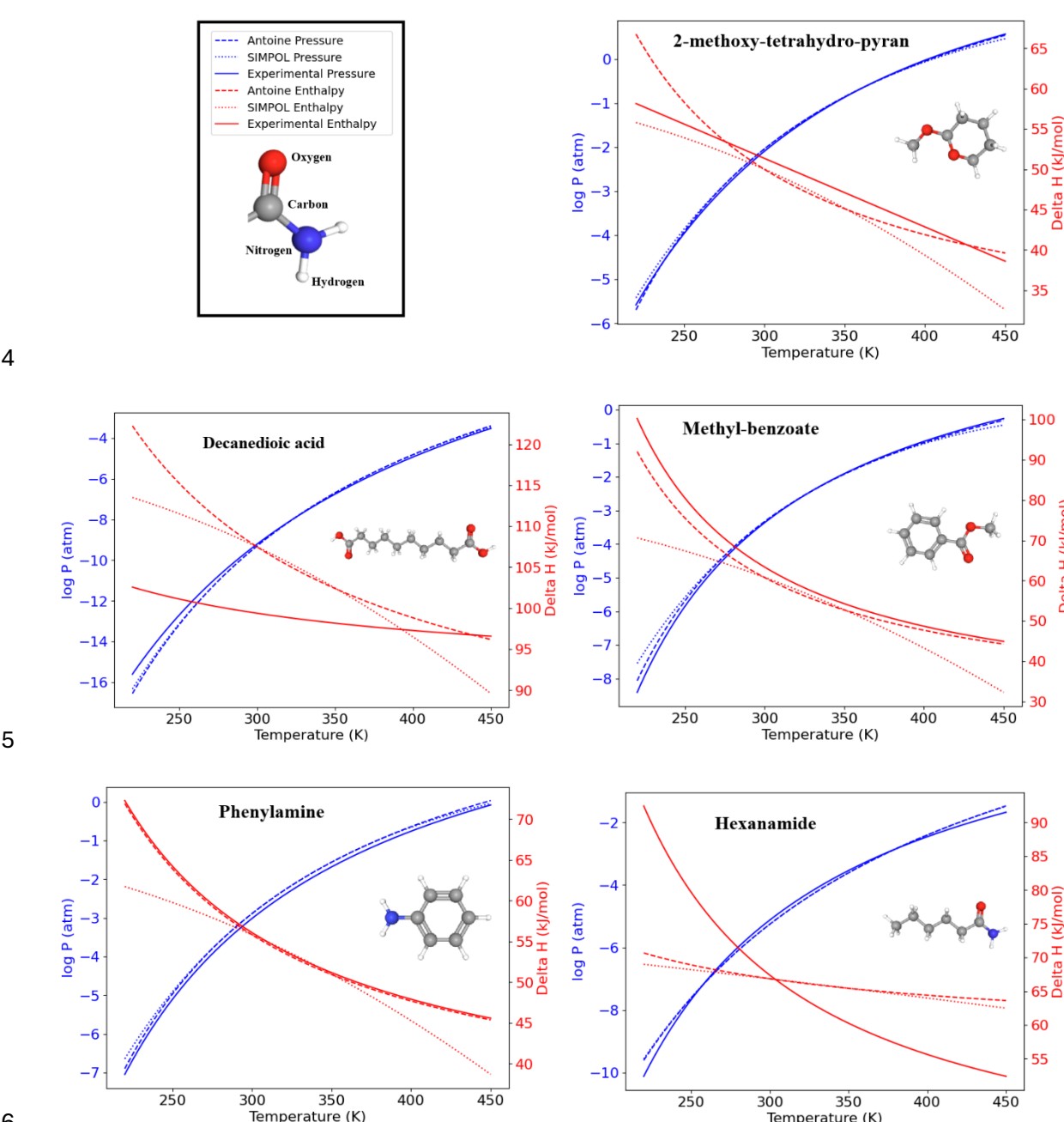

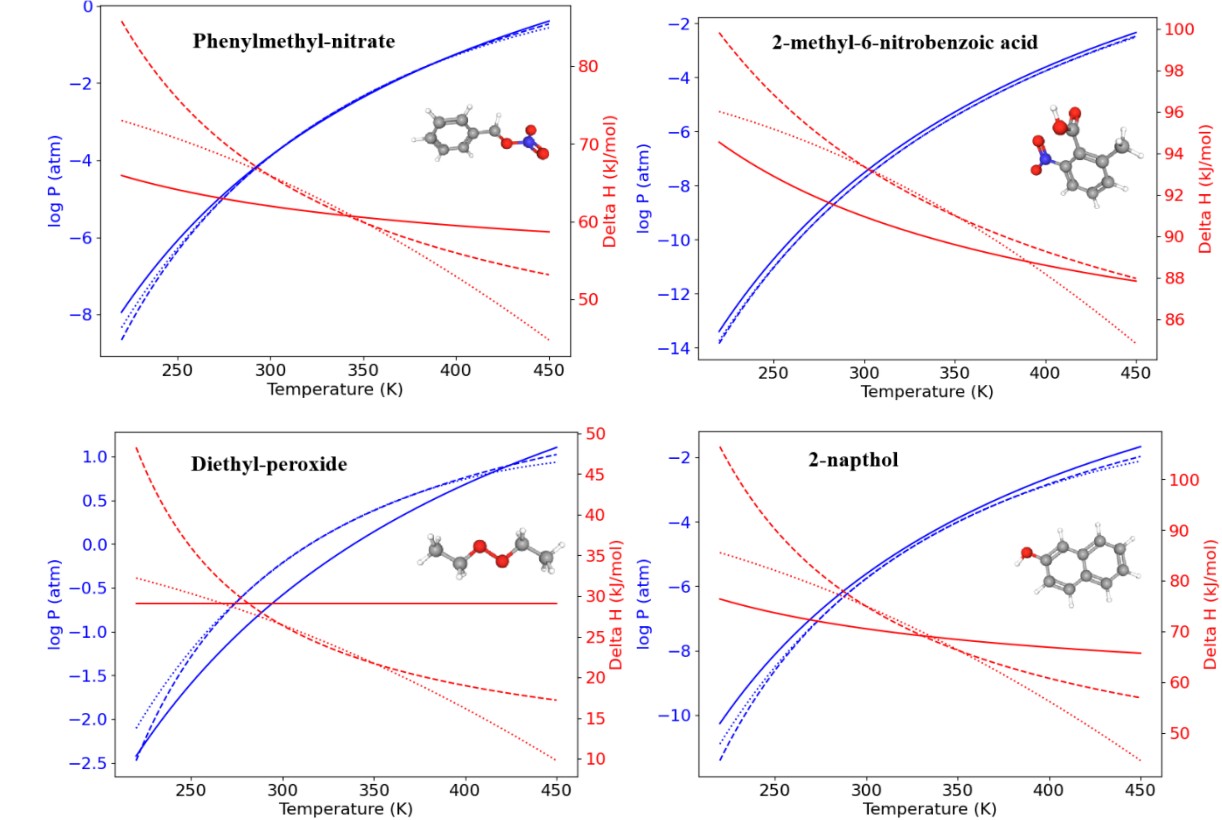

**Figure 7**. Temperature dependence of saturation vapor pressure and enthalpy of vaporization for nine representative organic compounds with nine distinct functional groups predicted by VaPOrS using the Antoine and SIMPOL equations. The left y-axis shows the logarithmic saturation vapor pressure (in atm), and the right y-axis displays the enthalpy of vaporization (in kJ/mol). The results demonstrate the data generated by Antoine and SIMPOL methods across the temperature range, with experimental data closely matching the theoretical predictions.

## 4. Results and discussion

Using VaPOrS, a detailed comparison was performed between the manual counting and calculation of functional groups and saturation vapor pressures and the automated results generated from the compounds' SMILES notations. The manual counting involved a systematic review of each compound's molecular structure, visually identifying and recording the functional groups, followed by calculating its saturation vapor pressure according to SIMPOL group contributions. This was then cross-referenced with the automated results generated by VaPOrS to ensure consistency. The procedure was performed in three steps described next.

### 4.1. MCM data

In the first step, a dataset of 126 primary VOCs sourced from the Master Chemical Mechanism
(MCM) database are evaluated. While the MCM database provides detailed chemical
mechanisms for atmospheric chemistry, its coverage of primary organic compounds is
relatively limited compared to the vast diversity of VOCs present in the atmosphere.
Nonetheless, it serves as a valuable resource for validation, given its detailed representation of
key compounds. Notably, the MCM database not only provides the structures of these
compounds but also includes their corresponding SMILES notation, facilitating an accurate
assessment of functional group presence through the VaPOrS method.

Table 1 presents a sample comparison between the manual and automated counts of the
functional groups for representative compounds from several categories in the MCM, including
Alcohols and Glycols, Aldehydes, Alkanes, Alkenes, Alkynes, Aromatics, Dialkenes, Esters,
Ethers and Glycol Ethers, Ketones, Monoterpenes and Sesquiterpenes, Organic Acids, and
Unclassified compounds. As shown, the results from both methods are in complete agreement,
with 0% discrepancy across all cases.

**Table 1:** Comparison of Manual and Automated Counts of Functional Groups for Representative Compounds across Various Categories in the MCM Database.

| Category | Compounds | SMILES | Functionals | Consistency |
|---|---|---|---|---|
| Alcohols and Glycols | CYCLOHEXANOL | OC1CCCCC1 | 6 carbons, 1 nonaromatic ring, 1 hydroxyl | 100% |
| Aldehydes | PROPENAL | C=CC=O | 3 carbons, 1 C=C (non-aromatic), 1 aldehyde | 100% |
| Alkanes | 3-METHYLPENTANE | CCC(C)CC | 6 carbons | 100% |
| Alkenes | 1-HEXENE | CCCCC=C | 6 carbons, 1 C=C (non-aromatic) | 100% |
| Alkynes | ETHYNE | C#C | 2 carbons | 100% |
| Aromatics | ETHYL BENZENE | CCc1ccccc1 | 8 carbons, 1 aromatic ring | 100% |
| Dialkenes | 1-3 BUTADIENE | C=CC=C | 4 carbons, 2 C=C (non-aromatic) | 100% |
| Esters | ETHYL ACETATE | CCOC(=O)C | 4 carbons, 1 ester | 100% |
| Ethers and Glycol Ethers | 2-METHOXY ETHANOL | COCCO | 3 carbons, 1 hydroxyl, 1 ether | 100% |
| Ketones | CYCLOHEXANONE | O=C1CCCCC1 | 6 carbons, 1 nonaromatic ring, 1 ketone | 100% |
| Monoterpenes and Sesquiterpenes | ALPHA-PINENE | CC1=CCC2CC1C2(C)C | 10 carbons, 2 nonaromatic rings, 1 C=C (non-aromatic), | 100% |
| Organic Acids | PROPANOIC ACID | CCC(=O)O | 3 carbons, 1 carboxylic acid | 100% |
| Unclassified | ETHYLENE OXIDE | O1CC1 | 2 carbons, 1 nonaromatic ring, 1 ether (alicyclic) | 100% |

In the second phase, alpha-pinene and benzene were selected as case studies to evaluate the

species formed during their tropospheric degradation via gas-phase chemical processes,

focusing on functional group occurrence and saturation vapor pressure. This analysis leveraged

the detailed mechanism in the MCM to further validate the automated functional group

detection system's accuracy in modeling atmospheric chemistry. For each species, the

occurrences of functional groups were manually counted, and saturation vapor pressure was

calculated using the SIMPOL method. Their SMILES notation was then input into the VaPOrS

to automatically obtain functional group counts and saturation vapor pressures. The saturation

vapor pressures obtained through both methods are compared in Figures 8 and 9 for alpha-

pinene and benzene, respectively, where the y-axis represents the logarithmic saturation vapor

pressure and the x-axis the molar mass of each species. Automated results are shown as color

bars based on the number of detected functional groups, and their manual counterparts are

displayed as points. The perfect alignment of points atop bars for each species indicates

excellent agreement between both approaches. It is worth mentioning that C6H6N2O11 and

CH2O (i.e., NNCATECOOH and HCHO in the MCM) were recognized as the least and most

volatile species in the benzene oxidation process. On the other hand, C9H16O6 and CH3O

(i.e., C922OOH and CH3O in the MCM) were the least and most volatile species in the alpha-

pinene oxidation process.

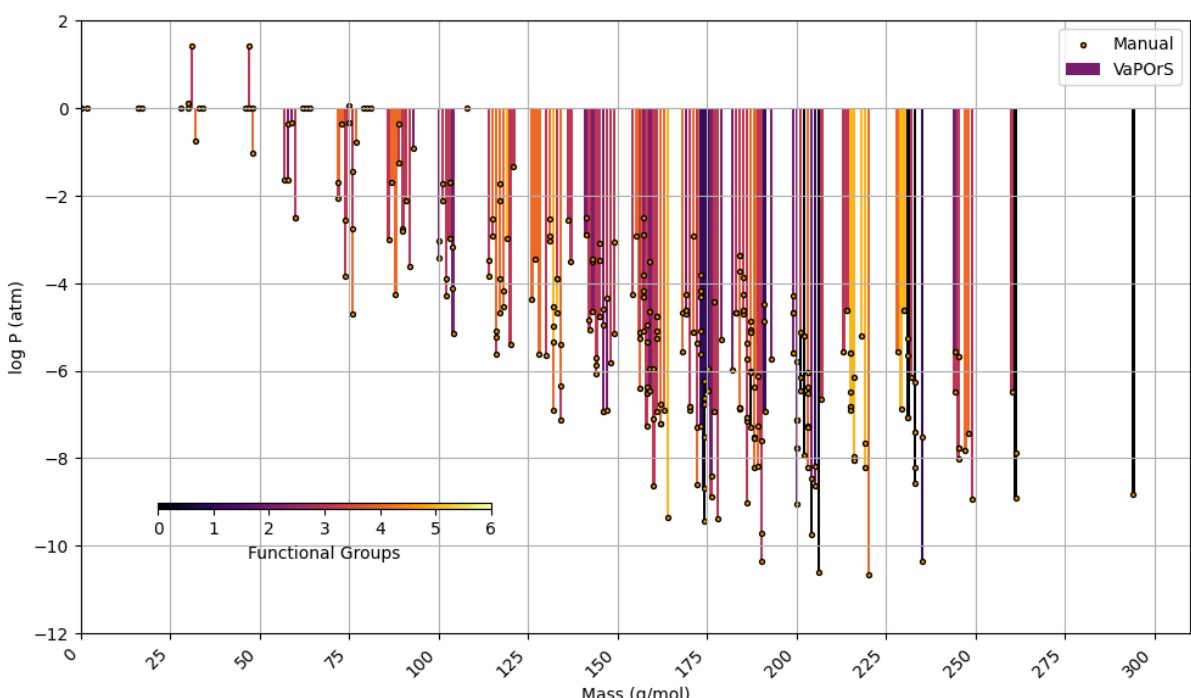

**Figure 8.** Comparison of saturation vapor pressure results for tropospheric degradation species of alpha-pinene.
The figure presents the log saturation vapor pressure versus molar mass for species formed from the
tropospheric oxidation of alpha-pinene according to MCM. Bars show results from the automated VaPOrS code,
with colors based on detected functional group number involved in the chemical structure of species, and points
reflect manually calculated values. The alignment of points atop bars demonstrates the perfect consistency
between automated and manual calculations.

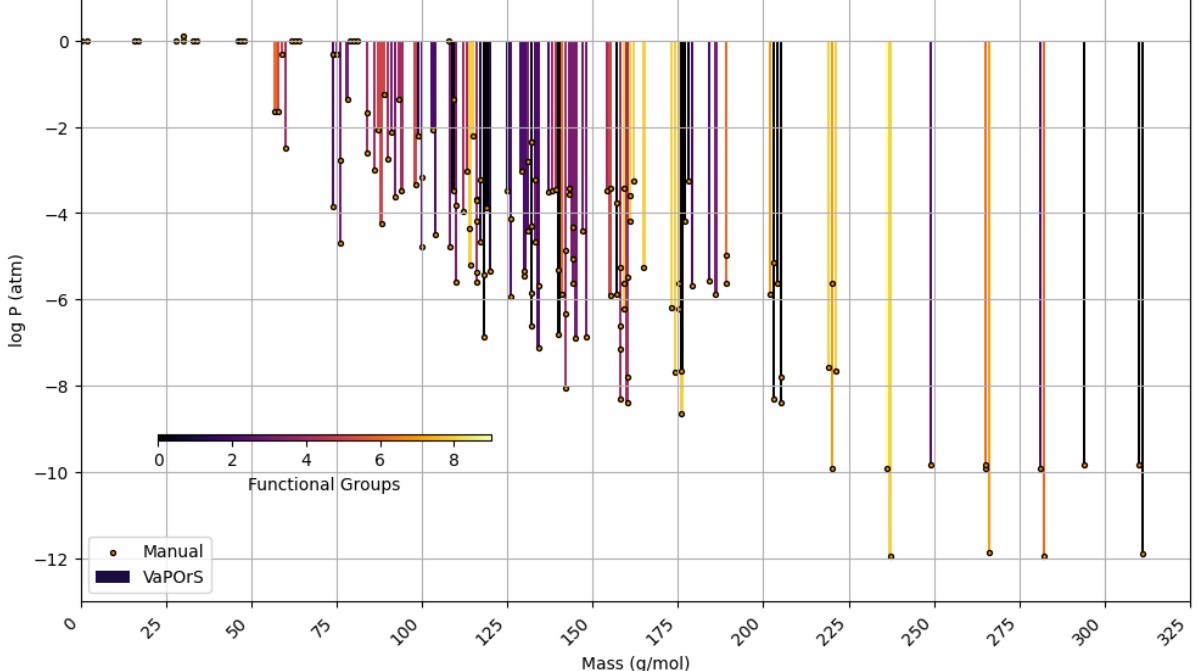

**Figure 9.** Comparison of saturation vapor pressure results for tropospheric degradation species of benzene. The
figure illustrates the log saturation vapor pressure versus molar mass for species derived from benzene
degradation according to MCM. The bars represent saturation vapor pressures calculated by the automated
VaPOrS code, with colors based on detected functional group number involved in the chemical structure of
species, while points indicate manually obtained values. The close alignment of points with bars highlights the
accuracy of the automated method relative to manual calculations.

**4.2.autoAPRAM-fw data**

In the third stage of this analysis, the VaPOrS model was utilized to determine the saturation

pressures of over 850 potential chemical species produced through the autoxidation of alkoxy

and peroxy radicals, which emerge during benzene degradation. The initial radicals were

defined by the MCM, with their respective saturation vapor pressures detailed in Figure 9. Conversely, the products of autoxidation were generated using the autoAPPRAM-fw tool (Pichelstorfer et al. 2024), with their potential structures represented by SMILES notation. Demonstrating its efficiency, VaPOrS analyzed all SMILES entries within a single second, accurately counting the required functional groups and calculating corresponding saturation vapor pressures. The results of these predictions are illustrated in Figure 10.

Figure 10 demonstrates the saturation vapor pressures achieved through the VaPOrS and compares them with their manually calculated counterparts. Automated results are shown as colorful bars based on the number of detected functional groups reaching as high as 15 for some autoAPRAMfw products, and manual results are depicted as points. The compatibility of points atop bars for each species indicates excellent agreement between both approaches.

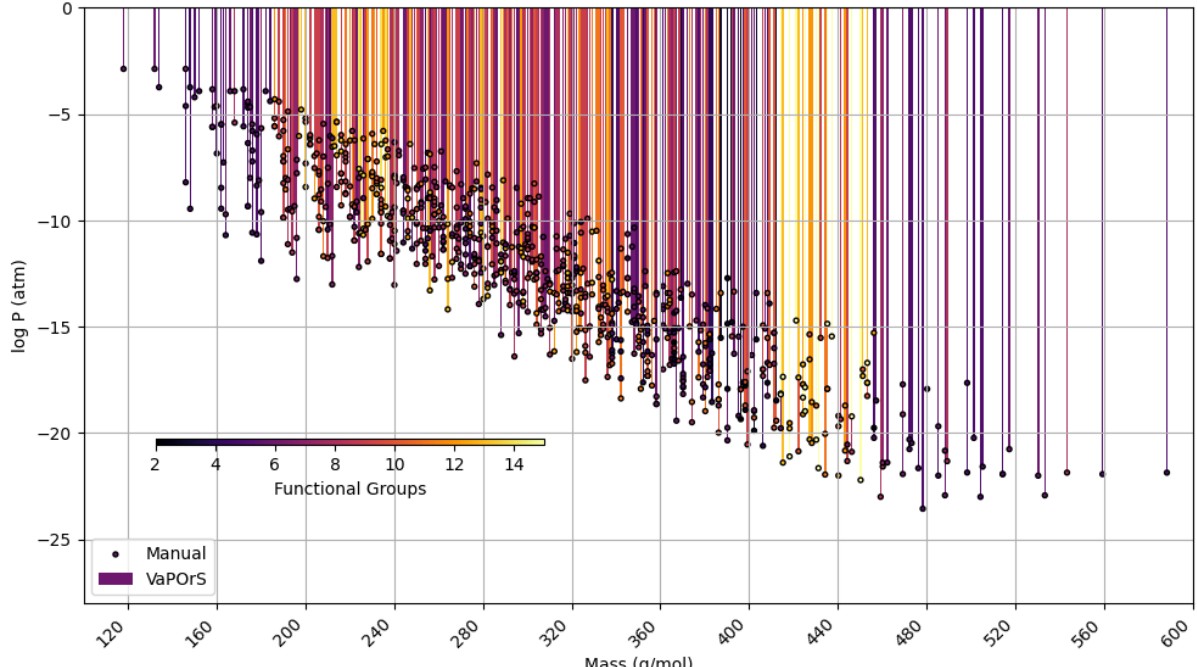

**Figure 10.** Comparison of saturation vapor pressure results for autoxidation species of benzene. The figure illustrates the log saturation vapor pressure versus molar mass for species derived from autoxidation of initial alkoxy and peroxy radicals of benzene degradation. The bars represent saturation vapor pressures calculated by the automated VaPOrS code, with colors based on detected functional group number involved in the chemical structure of species, while points indicate manually obtained values. The close alignment of points with bars highlights the accuracy of the automated method relative to manual calculations.

Figure 11 presents 2D and 3D scatter plots comparing the logarithmic saturation vapor pressures obtained from VaPOrS (log P_VaPOrS) with those predicted by the EVAPORATION (log P_EVAPORATION), Myrdal-Yalkowsky (log P_Myrdal_Yalkowsky), and Nanoolal (log P_Nanoolal) methods for autoxidation products. Each marker represents a compound, with color indicating the number of functional groups in its structure. The red dashed line in the 2D plots signifies the theoretical 1:1 relationship, where the saturation vapor pressures predicted by VaPOrS and the other models would be equivalent. Moreover, the 3D plots include the molar mass of species to give more details of the achieved results.

Further analysis shows that the current VaPOrS predictions based on SIMPOL parameterization align closely with those from the EVAPORATION method in the higher saturation vapor pressure range (approximately -6 to -15 on the logarithmic scale), with the EVAPORATION method tending to overestimate saturation vapor pressures for species with greater functional complexity. In contrast, VaPOrS demonstrates good agreement with the Nanoolal method in the lower saturation vapor pressure range (approximately -15 to -24), particularly for molecules with a high functional group count. The Myrdal-Yalkowsky method, however, totally overestimates saturation vapor pressures across the board compared to VaPOrS, with deviations increasing as functional group complexity rises. It is important to note that the current values are essentially SIMPOL-based predictions, so while these comparisons are informative, the general trends have been discussed in previous studies. However, as highlighted by (Isaacman-VanWertz and Aumont 2021), a combination of existing methods, potentially an average of them, has been suggested to yield the most reliable saturation vapor pressure estimates. Acknowledging this, a future refinement of the current approach could involve assessing whether incorporating such a hybrid method improves agreement with experimental data.

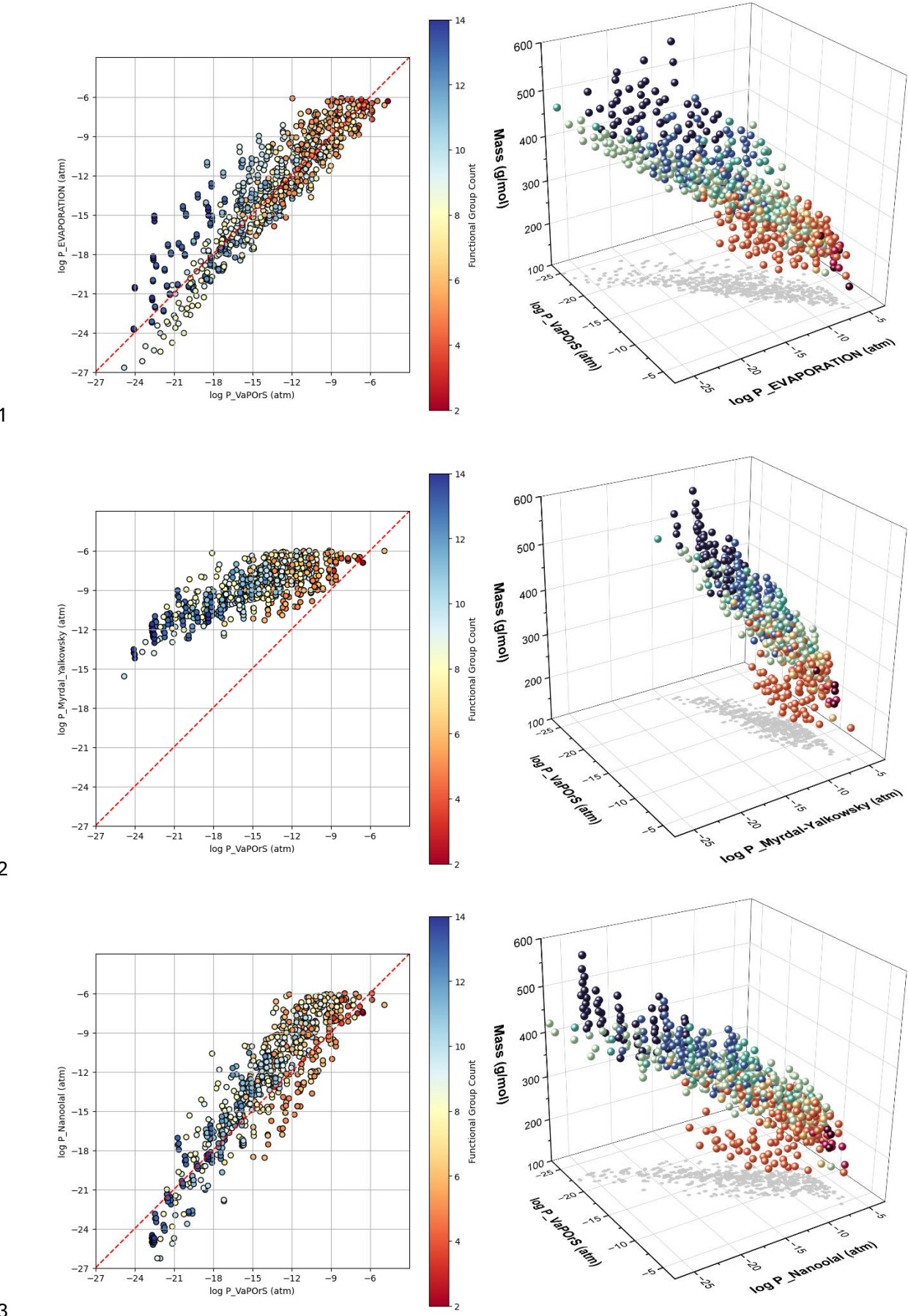

**Figure 11.** Comparison of logarithmic saturation vapor pressures predicted by the VaPOrS model (log P_VaPOrS) with those from the EVAPORATION (log P_EVAPORATION), Nanoolal (log P_Nanoolal), and Myrdal-Yalkowsky (log P_Myrdal_Yalkowsky) methods for autoxidation products derived from benzene degradation. Each data point represents a compound, with color indicating the functional group count. The red dashed line in the 2D plots represents the theoretical 1:1 relationship, where predictions from VaPOrS and other methods would be equivalent. The 3D plots include molar mass variation as well to give a comprehensive view of relationships between the parameters.

Using VaPOrS, the benzene-derived oxidation and autoxidation products were further classified into volatility categories based on their effective saturation mass concentration ($C^*$, $\mu g\ m^{-3}$). These include ultra-low-volatility organic compounds (ULVOC, $C^* \leq 3\times10^{-9}\ \mu g\ m^{-3}$), extremely low-volatility organic compounds (ELVOC, $3\times10^{-9} < C^* \leq 3\times10^{-5}\ \mu g\ m^{-3}$), low-volatility organic compounds (LVOC, $3\times10^{-5} < C^* \leq 3\times10^{-1}\ \mu g\ m^{-3}$), semi-volatile organic compounds (SVOC, $3\times10^{-1} < C^* \leq 3\times10^{2}\ \mu g\ m^{-3}$), and intermediate-volatility organic compounds (IVOC, $3\times10^{2} < C^* \leq 3\times10^{6}\ \mu g\ m^{-3}$) (Simon et al. 2020).

Figure 12 presents the relationship between molar mass and effective saturation concentration for these products. Oxidation products from the MCM and autoxidation products from autoAPPRAM-fw were both introduced into the VaPOrS framework to calculate their saturation concentrations. Each data point corresponds to an individual compound. The regression line highlights a strong negative correlation between molar mass and vapor pressure, indicating that molecular growth shifts compounds toward the low-volatility regime. The inset pie chart shows the molar-mass-weighted distribution across volatility classes, underlining the prevalence of ELVOC and ULVOC species. These results demonstrate that autoxidation significantly enhances the formation of low-volatility vapors, which play a key role in secondary organic aerosol (SOA) formation.

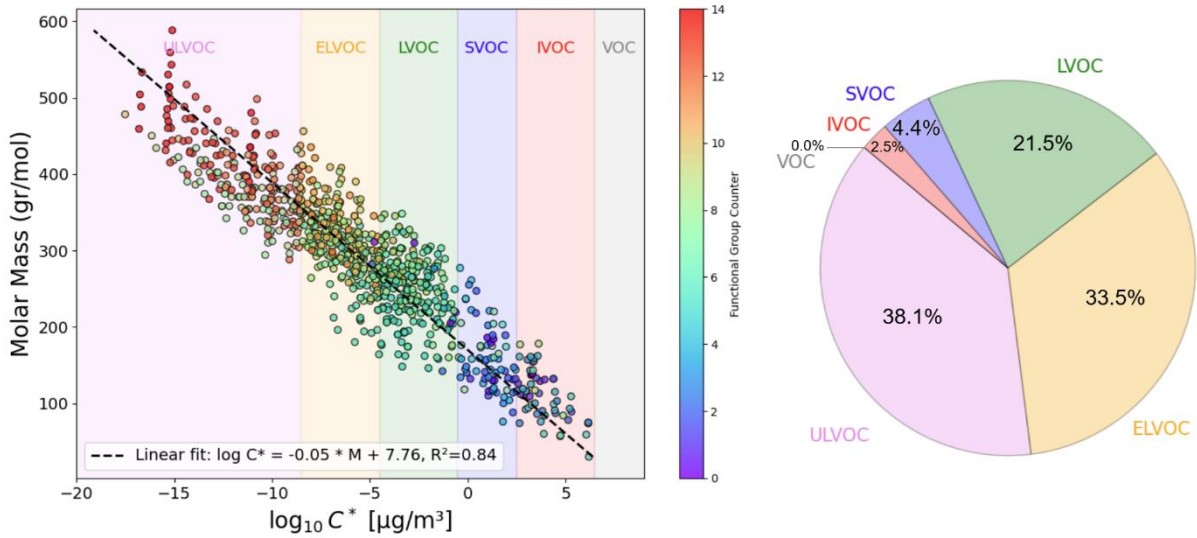

**Figure 12.** Relationship between molar mass and effective saturation concentration (log C*, µg m⁻³) of benzene-derived products. Oxidation products (MCM) and autoxidation products (autoAPPRAM-fw) were both introduced into VaPOrS for the calculation of their saturation concentrations. Each point represents an individual compound. The dashed line denotes the linear regression fit, showing a strong negative correlation between molar mass and vapor pressure. Volatility classes are shown as shaded regions: ULVOC (C* ≤ 3×10⁻⁹ µg m⁻³), ELVOC (3×10⁻⁹ < C* ≤ 3×10⁻⁵ µg m⁻³), LVOC (3×10⁻⁵ < C* ≤ 3×10⁻¹ µg m⁻³), SVOC (3×10⁻¹ < C* ≤ 3×10² µg m⁻³), and IVOC (3×10² < C* ≤ 3×10⁶ µg m⁻³). The inset pie chart illustrates the molar-mass-weighted fractions of compounds in each volatility category, highlighting the dominance of ELVOC and ULVOC species.

## 4.3. Benchmarking against existing tools

While UManSysProp also employs SMILES notations for vapor pressure estimation, our work was motivated by its repeated and verifiable failures to correctly identify functional groups across a range of organic species. Such misclassifications compromise vapor pressure estimation, particularly for chemically complex multifunctional compounds relevant to atmospheric oxidation and SOA formation. To quantify this, we benchmarked UManSysProp and VaPOrS against the original compound set used in the SIMPOL development paper and validated in section 4. UManSysProp frequently misidentified functional groups listed in Table 2, resulting in deviations from SIMPOL values, whereas VaPOrS consistently reproduced the expected outputs.

**Table 2.** Benchmark of UManSysProp and VaPOrS against the original SIMPOL compounds from Pankow and Asher (2008). Several functional groups critical for vapor pressure estimation were missed by UManSysProp,

1     resulting in deviations from SIMPOL values. VaPOrS accurately reproduced SIMPOL predictions in all cases.

2                        All vapor pressure values are expressed as log p (in atm).

| Compound | SMILES | SIMPOL | UManSysProp | VaPOrS | Experimental |
|---|---|---|---|---|---|
| formamide | C(=O)N | -2.6493 | 1.8307 | -2.6493 | -3.1778 |
| ethyl-formamide | CCNC=O | -3.2014 | 1.1292 | -3.2014 | -3.0353 |
| methyl-formamide | CNC=O | -2.8507 | 1.4799 | -2.8507 | -2.6507 |
| diethyl-formamide | CCN(CC)C=O | -2.0970 | 0.4278 | -2.0970 | -1.8235 |
| dimethyl-formamide | CN(C)C=O | -1.3956 | 1.1292 | -1.3956 | -1.4152 |
| dimethyl-hydroxylamine | CN(C)O | -1.1686 | 1.4799 | -1.1686 | -0.7208 |
| n-butyl-benzoate | CCCCOC(=O)c1ccccc1 | -3.3742 | -2.2884 | -3.3742 | -3.4300 |
| 2-methyl-propyl-benzoate | CC(C)COC(=O)c1ccccc1 | -3.3742 | -2.2884 | -3.3742 | -2.9881 |
| n-propyl-benzoate | CCCOC(=O)c1ccccc1 | -3.0235 | -1.9377 | -3.0235 | -2.7297 |
| ethyl-benzoate | CCOC(=O)c1ccccc1 | -2.6728 | -1.5870 | -2.6728 | -2.5309 |
| methyl-benzoate | COC(=O)c1ccccc1 | -2.3220 | -1.2363 | -2.3220 | -2.3216 |
| acetic-acid,-phenyl-ester | CC(=O)Oc1ccccc1 | -2.3220 | -2.1088 | -2.3220 | -2.2995 |
| dimethyl-1,2-benzenedicarboxylate | COC(=O)c1ccccc1C(=O)OC | -4.1093 | -1.9377 | -4.1093 | -4.2168 |
| dimethyl-benzene-1,3-dicarboxylate | COC(=O)c1cccc(c1)C(=O)OC | -4.1093 | -1.9377 | -4.1093 | -4.1260 |
| dimethyl-benzene-1,4-dicarboxylate | COC(=O)c1ccc(cc1)C(=O)OC | -4.1093 | -1.9377 | -4.1093 | -4.0839 |
| di-n-butyl-ethanedicarboxylate | CCCCOC(=O)C(=O)OCCCC | -3.4973 | -1.3257 | -3.4973 | -3.3166 |
| diethyl-ethanedicarboxylate | CCOC(=O)C=CC(=O)OCC | -2.7959 | -0.7558 | -2.7959 | -2.5070 |
| ethyl-2-nitropropionate | CCOC(=O)C(C)[N+](=O)[O-] | -3.3118 | -2.4828 | -3.3118 | -2.6618 |
| methyl-2-nitro-propionate | CC(C(=O)OC)[N+](=O)[O-] | -2.9611 | -2.1320 | -2.9611 | -2.6133 |

Further comparison was carried out for the 126 primary VOCs from MCM database used in

section 4.1. UManSysProp yielded incorrect predictions for at least four representative

compounds listed in Table 3, again due to functional group misidentification, while VaPOrS

produced accurate results.

**Table 3.** Evaluation of four primary VOCs from the MCM database. UManSysProp produced incorrect vapor

pressure values due to the misidentification of functional groups, while VaPOrS matched SIMPOL outputs

precisely. All vapor pressure values are expressed as log p (in atm).

| Compound | SMILES | SIMPOL | UManSysProp | VaPOrS |
|---|---|---|---|---|
| formaldehyde | C=O | 0.6863 | 1.8307 | 0.6863 |

| | | | | |
|---|---|---|---|---|
| formic acid | OC=O | -1.2397 | 1.8307 | -1.2397 |
| methyl ester | COC=O | 0.3942 | 1.4799 | 0.3942 |
| BCARY | C/C1=C/CCC(=C)C2CC(C)(C)C2CC\1 | -3.5640 | -3.4531 | -3.5640 |

The discrepancies were even more pronounced for multifunctional oxidation products. For

instance, for the α-pinene and benzene oxidation products used in section 4.1, UManSysProp

failed to correctly compute vapor pressures for at least 20 products of each precursor, listed in

Tables 4 and 5. VaPOrS, in contrast, successfully identified all functional groups and aligned

with SIMPOL calculations in every case.

**Table 4.** Comparison of predicted vapor pressures for 20 α-pinene oxidation products. UManSysProp failed to

recognize various multifunctional and peroxide-containing groups, leading to considerable errors. VaPOrS

correctly identified all functional groups and reproduced the SIMPOL values. All vapor pressure values are

expressed as log p (in atm).

| SMILES | SIMPOL | UManSysProp | VaPOrS |
|---|---|---|---|
| [O]OC(=O)CC(=O)C=O | -2.2520 | -1.3089 | -2.2520 |
| [O]OC(=O)C1CC(C(=O)O)C1(C)C | -4.7488 | -3.8057 | -4.7488 |
| [O]OC(=O)CC1CC(C(=O)O)C1(C)C | -5.0995 | -4.1564 | -5.0995 |
| [O]OC(=O)C1CC(C(=O)C)C1(C)C | -2.9721 | -2.0290 | -2.9721 |
| O=CCC1CC(C(=O)[O])C1(C)C | -3.1733 | -2.2302 | -3.1733 |
| O=CCC1CC(C(=O)O[O])C1(C)C | -3.1733 | -2.2302 | -3.1733 |
| [O]OC(=O)CC1CC(C(=O)CO)C1(C)C | -5.1871 | -4.2440 | -5.1871 |
| [O]OC(=O)CC1CC(C(=O)C)C1(C)C | -3.3229 | -2.3797 | -3.3229 |
| CC(=O)O[O] | 0.5368 | 1.4799 | 0.5368 |
| [O]OC(=O)CC(=O)CC=O | -2.6027 | -1.6596 | -2.6027 |
| [O]OC(=O)CC(=O)CC(=O)C(=O)C | -4.0461 | -3.1030 | -4.0461 |
| [O]OC(=O)CC(=O)C(=O)C | -2.4015 | -1.4584 | -2.4015 |
| CC(=O)C(O)C(=O)O[O] | -2.972 | -2.0288 | -2.972 |
| [O]OC(=O)CC(=O)C(=O)CO | -4.2658 | -3.3227 | -4.2658 |
| [O]OC(=O)CC(=O)CC(O)C(=O)C | -4.9673 | -4.0241 | -4.9673 |
| CC(=O)C(O)CC(=O)O[O] | -3.3227 | -2.3795 | -3.3227 |
| [O]OC(=O)CC=O | -0.9581 | -0.0150 | -0.9581 |
| [O]OC(=O)C=O | -0.6074 | 0.3356 | -0.6074 |
| OCC(=O)O[O] | -1.3274 | -0.3842 | -1.3274 |
| O=CC(=O)CC(=O)C(C)(ON(=O)=O)C(=O)O[O] | -6.1581 | -5.2149 | -6.1581 |

**Table 5.** Predicted vapor pressures of 20 benzene oxidation products. UManSysProp produced erroneous outputs due to functional group detection errors, particularly in conjugated and peroxide-bearing species. VaPOrS successfully identified all necessary groups and aligned with SIMPOL calculations. All vapor pressure values are expressed as log p (in atm).

| SMILES | SIMPOL | UManSysProp | VaPOrS |
|---|---|---|---|
| [O]OC(=O)C1OC1C=CC=O | -2.7960 | -1.8529 | -2.7960 |
| OC1COC(=O)C1=O | -3.2255 | -2.1397 | -3.2255 |
| C1OC(=O)C=C1 | -0.9785 | 0.1072 | -0.9785 |
| [O]OC(=O)C=CC(=O)C=O | -2.7342 | -1.7911 | -2.7342 |
| OC(C=O)C(=O)C=CC(=O)O[O] | -4.9492 | -4.0061 | -4.9492 |
| O=CCOC(=O)C=O | -2.5958 | -1.5100 | -2.5958 |
| O=CCOC(=O)C(=O)O | -4.5220 | -3.4362 | -4.5220 |
| [O]OC(=O)C1OC1C=O | -1.9631 | -1.0200 | -1.9631 |
| [O]OC(=O)C=O | -0.6074 | 0.3356 | -0.6074 |
| OC(C=O)C(=O)O[O] | -2.8224 | -1.8793 | -2.8224 |
| OCC(=O)O[O] | -1.3274 | -0.3842 | -1.3274 |
| O=CC=CC(=O)[O] | -1.4403 | -0.4972 | -1.4403 |
| [O]OC(=O)C=CC=O | -1.4403 | -0.4972 | -1.4403 |
| O=CC(=CC(=O)[O])N(=O)=O | -3.2652 | -2.3220 | -3.2652 |
| O=N(=O)C12OOC(C2O)C(O)([O])C(=C1)O | -8.2564 | -6.3921 | -8.2564 |
| [O]OC1(O)C(=CC2(OOC1C2O)N(=O)=O)O | -8.2564 | -6.3921 | -8.2564 |
| OOC1(O)C(=CC2(OOC1C2O)N(=O)=O)O | -10.3826 | -8.5183 | -10.3826 |
| O=N(=O)OC1C2OOC1(C=C(O)C2([O])O)N(=O)=O | -8.3029 | -6.4386 | -8.3029 |
| [O]OC1(O)C(=CC2(OOC1C2ON(=O)=O)N(=O)=O)O | -8.3029 | -6.4386 | -8.3029 |
| OOC1(O)C(=CC2(OOC1C2ON(=O)=O)N(=O)=O)O | -10.4291 | -8.5648 | -10.4291 |

Finally, we evaluated 180 oxidation products of aromatic carbonyls obtained from high-resolution mass spectrometry studies (Barua et al. 2025). UManSysProp produced erroneous values for 67 of these compounds, while VaPOrS correctly handled all cases, listed in Table 6.

**Table 6.** Analysis of oxidation products from OH-initiated autooxidation of aromatic carbonyls under different NOx conditions. UManSysProp failed in 67 cases due to group detection limitations. VaPOrS handled all compounds correctly via explicit SMILES-based pattern recognition. All vapor pressure values are expressed as log p (in atm).

| SMILES | SIMPOL | UManSysProp | VaPOrS |
|---|---|---|---|

| | | | |
|---|---|---|---|
| CC (=O) C (=O) C (O) =CC (=O) C (O) C=O | -8.4581 | -6.5938 | -8.4581 |
| CC (=O) C (=O) C (O) =CC (=O) C (OO) C=O | -8.7200 | -6.8557 | -8.7200 |
| CC (=O) C (=O) C (O) =CC (OO) C (=O) C=O | -8.7200 | -6.8557 | -8.7200 |
| CC (=O) C (=O) C (O) =CC (O [O] ) C (O) C=O | -7.5149 | -5.6506 | -7.5149 |
| CC (=O) C (=O) C (O) =CC (ON (=O) =O) C (O) C=O | -9.4257 | -7.5614 | -9.4257 |
| CC (=O) C (=O) C (O) =CC (OO) C (O) C=O | -9.6411 | -7.7768 | -9.6411 |
| CC (=O) C (=O) C (O) =CC (O) C (OO) C=O | -9.6411 | -7.7768 | -9.6411 |
| CC (=O) C (=O) C (O) =CC (=O) C (O) C (=O) OO | -9.4298 | -7.5655 | -9.4298 |
| CC (=O) C (=O) C (O) =CC (O [O] ) C (OO) C=O | -7.7768 | -5.9126 | -7.7768 |
| CC (=O) C (=O) C (O) C (OO) C=CC (O [O] ) =O | -7.5756 | -6.6325 | -7.5756 |
| CC (=O) C (=O) C (O) =CC (ON (=O) =O) C (OO) C=O | -9.6876 | -7.8233 | -9.6876 |
| CC (=O) C (=O) C (O) =CC (O) C (O) C (=O) OO | -10.351 | -8.4866 | -10.351 |
| CC (=O) C (=O) C (O) =CC (=O) C (OO) C (=O) OO | -9.6917 | -7.8274 | -9.6917 |
| CC (=O) C (=O) C (O) =CC (O [O] ) C (O) C (=O) OO | -8.4866 | -6.6223 | -8.4866 |
| CC (=O) C (=O) C (O) =CC (ON (=O) =O) C (O) C (=O) OO | -10.397 | -8.5331 | -10.397 |
| CC (=O) C (=O) C (O) =CC (OO) C (OO) C (=O) O [O] | -9.7018 | -6.8944 | -9.7018 |
| CC (=O) C (=O) C (O) =CC (O [O] ) C (OO) C (=O) OO | -8.7485 | -6.8843 | -8.7485 |
| CC (=O) C (=O) C (O) =CC (OO) C (OO) C (=O) ON (=O) =O | -10.669 | -8.8052 | -10.669 |
| CC (=O) C (=O) C (O) =CC (ON (=O) =O) C (OO) C (=O) OO | -10.659 | -8.7950 | -10.659 |
| CC (=O) C (=O) C (O) =CC (OO) C (OO) C (=O) OO | -10.874 | -9.0105 | -10.874 |
| [O] OCC (=O) C (=O) C (O) =CC (OO) C (O) C (OO) =O | -10.612 | -8.7485 | -10.612 |
| O=N (=O) OCC (=O) C (=O) C (O) =CC (OO) C (O) C (OO) =O | -12.523 | -10.659 | -12.523 |
| O=C (C=O) C (O) =CC (=O) C (O) C=O | -8.3085 | -6.4442 | -8.3085 |
| O=C (C=O) C (O) =CC (=O) C (OO) C=O | -8.5704 | -6.7062 | -8.5704 |
| O=C (C=O) C (O) =CC (OO) C (=O) C=O | -8.5704 | -6.7062 | -8.5704 |
| O=C (C=O) C (O) =CC (O [O] ) C (O) C=O | -7.3654 | -5.5011 | -7.3654 |
| O=C (C=O) C (O) =CC (O) C (OO) C=O | -9.4916 | -7.6273 | -9.4916 |
| O=C (C=O) C (O) =CC (=O) C (O) OO | -8.9397 | -7.0754 | -8.9397 |
| O=C (C=O) C (O) =CC (=O) C (O) C (=O) OO | -9.2802 | -7.4159 | -9.2802 |
| O=C (C=O) C (O) =CC (O [O] ) C (OO) C=O | -7.6273 | -5.7630 | -7.6273 |
| O=C (C=O) C (O) =CC (OO) C (OO) C=O | -9.7535 | -7.8892 | -9.7535 |
| O=C (C=O) C (O) =CC (=O) C (OO) C (=O) OO | -9.5421 | -7.6779 | -9.5421 |
| O=C (C=O) C (O) =CC (O [O] ) C (O) C (=O) OO | -8.3371 | -6.4728 | -8.3371 |
| O=C (C=O) C (O) =CC (O [O] ) C (OO) C (=O) OO | -8.5990 | -6.7347 | -8.5990 |
| O=C (C=O) C (O) =CC (OO) C (OO) C (=O) OO | -10.725 | -8.8609 | -10.725 |
| O=CCC (=O) C (O) =CC (=O) C (O) C=O | -8.659 | -6.7950 | -8.659 |
| O=CCC (=O) C (O) =CC (=O) C (OO) C=O | -8.9212 | -7.0569 | -8.9212 |
| O=CCC (=O) C (O) =CC (OO) C (=O) C=O | -8.9212 | -7.0569 | -8.9212 |
| O=CCC (=O) C (O) =CC (O [O] ) C (O) C=O | -7.7161 | -5.8518 | -7.7161 |
| O=CCC (=O) C (O) =CC (ON (=O) =O) C (O) C=O | -9.6269 | -7.7626 | -9.6269 |
| O=CCC (=O) C (O) =CC (O) C (OO) C=O | -9.8423 | -7.9780 | -9.8423 |
| O=CCC (=O) C (O) =CC (=O) C (O) C (=O) OO | -9.631 | -7.7667 | -9.631 |
| O=CCC (=O) C (O) =CC (O [O] ) C (OO) C=O | -7.9780 | -6.1137 | -7.9780 |

| | | | |
|---|---|---|---|
| O=CCC(=O)C(O)=CC(ON(=O)=O)C(OO)C=O | -9.8888 | -8.0245 | -9.8888 |
| O=CCC(=O)C(O)=CC(OO)C(OO)C=O | -10.104 | -8.2399 | -10.104 |
| O=CCC(=O)C(O)=CC(O[O])C(O)C(=O)OO | -8.6878 | -6.8235 | -8.6878 |
| O=CCC(=O)C(O)=CC(ON(=O)=O)C(O)C(=O)OO | -10.598 | -8.7343 | -10.598 |
| O=CC(=O)C(=O)C(O)=CC(OO)C(O)C(=O)OO | -11.757 | -9.8929 | -11.757 |
| O=CCC(=O)C(O)=CC(OO)C(OO)C(=O)O[O] | -9.9030 | -7.0956 | -9.9030 |
| [O]OC(C=O)C(=O)C(O)=CC(OO)C(OO)C=O | -10.104 | -8.2399 | -10.104 |
| O=CCC(=O)C(O)=CC(O[O])C(OO)C(OO)=O | -8.9497 | -7.0854 | -8.9497 |
| O=CCC(=O)C(O)=CC(OO)C(OO)C(=O)ON(=O)=O | -10.870 | -9.0064 | -10.870 |
| O=N(=O)OC(C=O)C(=O)C(O)=CC(OO)C(OO)C=O | -12.015 | -10.150 | -12.015 |
| O=CCC(=O)C(O)=CC(ON(=O)=O)C(OO)C(OO)=O | -10.860 | -8.9962 | -10.860 |
| O=CCC(=O)C(O)=CC(OO)C(OO)C(=O)OO | -11.076 | -9.2116 | -11.076 |
| O=CC(OO)C(=O)C(O)=CC(OO)C(OO)C=O | -12.230 | -10.366 | -12.230 |
| O=CC(O[O])C(=O)C(O)=CC(OO)C(O)C(=O)OO | -10.814 | -8.9497 | -10.814 |
| O=CCC(=O)C(=O)C(OO)C(OO)C(O)C(=O)O[O] | -10.714 | -9.7716 | -10.714 |
| O=CC(ON(=O)=O)C(=O)C(O)=CC(OO)C(O)C(=O)OO | -12.724 | -10.860 | -12.724 |
| [O]OC(=O)C(OO)C(=O)C(O)=CC(OO)C(OO)C=O | -12.029 | -9.2218 | -12.029 |
| O=CCC(=O)C(=O)C(OO)C(OO)C(OO)C(=O)O[O] | -10.976 | -10.033 | -10.976 |
| O=N(=O)OC(=O)C(OO)C(=O)C(O)=CC(OO)C(OO)C=O | -12.996 | -11.132 | -12.996 |
| OOC(=O)C(OO)C(=O)C(O)=CC(OO)C(OO)C=O | -13.202 | -11.337 | -13.202 |
| O=CCC(=O)C(O)=CC(C(O)C=O)OOOC(C=O)C=CC(O)C(=O)CC=O | -15.749 | -13.885 | -15.749 |
| O=CCC(=O)C(O)=CC(C(O)C=O)OOOC1C=CC2OOC1C2(O)CC=O | -14.446 | -12.582 | -14.446 |
| O=CCC(=O)C(O)=CC(C(O)C=O)OOC(C(O)C=O)C=C(O)C(=O)CC=O | -18.176 | -14.447 | -18.176 |
| O=CCC(=O)C(O)=CC(C(OO)C=O)OOC(C(OO)C=O)C=C(O)C(=O)CC=O | -18.700 | -14.971 | -18.700 |

These results demonstrate systemic limitations of UManSysProp when applied to chemically diverse and rapidly expanding databases of atmospheric oxidation products. With the continuous introduction of new molecules into atmospheric models, a tool like VaPOrS, relying on explicit SMILES-based pattern recognition, offers a reliable, transparent, and scalable alternative for functional group detection and vapor pressure estimation.

The versatility of VaPOrS lies in both its algorithmic design and its potential for seamless integration into a wide range of atmospheric models. By directly parsing SMILES strings and identifying functional groups without relying on external libraries such as OpenBabel, VaPOrS provides a transparent, lightweight, and easily modifiable framework. This not only reduces dependency and installation challenges common to existing tools like UManSysProp but also ensures that the group detection logic remains fully auditable and extendable to new parameterization schemes. The validation results presented in this study confirm that VaPOrS correctly identifies functional groups across large external datasets such as the MCM and auto-

APRAM-fw, demonstrating both its robustness and its suitability for high-throughput applications.

Several widely used atmospheric models stand to benefit from the integration of VaPOrS. For example, the MCM, with its extensive SMILES database of VOCs, can directly leverage VaPOrS to automate the generation of temperature-dependent saturation vapor pressure equations. This automation ensures uniformity in property predictions, which is critical for large-scale models to simulate chemical reactions and transport processes consistently (Saunders et al. 2003; Jenkin et al. 2003). Likewise, MEGAN (Guenther et al. 2006) can utilize VaPOrS to improve biogenic VOC emission estimates by rapidly supplying vapor pressure and enthalpy of vaporization values, while models such as LOTOS-EUROS (Schaap et al. 2008), GEOS-Chem (Bey et al. 2001), and WRF-Chem (Grell et al. 2005) can exploit its scalability to process thousands of compounds efficiently. Regional models such as CMAQ (Byun and Schere 2006) and process-level models such as ADCHAM and ADCHEM (Roldin et al. 2014; 2011) can also benefit from VaPOrS's capability to refine gas-particle partitioning, aerosol growth rates, and ultimately climate-relevant feedbacks.

At the same time, it is important to recognize the limitations of group contribution approaches such as SIMPOL, which underpins this first implementation of VaPOrS. While reliable for many compounds, predictive accuracy declines as molecular complexity increases. Highly functionalized molecules may exhibit non-additive effects, such as steric hindrance, intramolecular hydrogen bonding, or electronic interactions, that are not captured by simple group summation rules. Previous studies have shown that such effects can dampen or amplify volatility changes in ways not accounted for by group contribution approaches. Another important limitation relates to tautomerism, in which a compound can exist in multiple chemically equivalent but structurally distinct forms, for example keto-enol tautomerism. These forms contain different functional groups. For instance, a keto tautomer may contain a carbonyl group (C=O), while its enol counterpart may contain one hydroxyl (OH) and one carbon-carbon double bond (C=C). Since functional group-based methods like SIMPOL assign unique parameters to each group, different tautomeric SMILES representations of the same compound can yield different vapor pressure predictions, despite the compound having a single experimentally measurable vapor pressure. VaPOrS does not attempt to canonicalize or normalize tautomeric forms; it faithfully parses and counts groups in the SMILES provided by the user or database. Thus, this ambiguity arises from the group-contribution framework itself rather than from VaPOrS. In practice, most studies adopt a convention of selecting the

thermodynamically more stable tautomer, commonly the keto form in the gas phase, as the reference structure, although this is not universally standardized. Should future investigations establish robust strategies for handling tautomerism, for example by developing distinct parameters for different tautomeric states, such improvements could be readily integrated into VaPOrS. Addressing these limitations, whether through correction terms, hybridization with other frameworks, or leveraging data-driven approaches such as machine learning, represents an important avenue for future development.

## 5.  Conclusion

This study introduced VaPOrS (Vapor Pressure in Organics via SMILES), a Python-based computational tool designed to automate the identification and quantification of functional groups in organic compounds and to calculate their saturation vapor pressures using the SIMPOL method. Three key contributions were demonstrated: (i) systematic detection of 30 functional groups relevant to saturation vapor pressure parameterization, (ii) automated computation of saturation vapor pressure and enthalpy of vaporization, and (iii) derivation of temperature-dependent relationships connecting these quantities. While the present work focused on SIMPOL, the underlying framework of VaPOrS is flexible and can be extended to other structure-based approaches, including group additivity and volatility basis set (VBS) methods. Its design also makes it readily adaptable for estimating additional thermodynamic properties beyond vapor pressure, broadening its utility in atmospheric and environmental chemistry.

The tool was rigorously validated against both manually curated functional group counts and external datasets. Across more than 1,000 compounds from the Master Chemical Mechanism (MCM) database and the auto-APRAM-fw autoxidation scheme, VaPOrS achieved perfect agreement in functional group recognition and produced saturation vapor pressure predictions consistent with SIMPOL and experimental data. Comparative benchmarking further showed that VaPOrS overcomes limitations of existing tools, such as UManSysProp, by avoiding misidentification of complex multifunctional groups and enabling transparent, modifiable detection logic. By combining accuracy, scalability, and adaptability, VaPOrS provides a robust and efficient solution for processing the vast chemical space of volatile organic compounds in large-scale atmospheric models. Its integration into frameworks targeting secondary organic aerosol formation and related processes without overheads can substantially accelerate simulations while improving reliability. Future work will expand the functional

group library, incorporate alternative parameterizations, and optimize computational performance, further extending the role of VaPOrS as a versatile tool for atmospheric chemistry and beyond.

**Data availability**

The raw data supporting the figures in the manuscript are openly available on Zenodo at: https://doi.org/10.5281/zenodo.15688105.

**Code availability**

The VaPOrS code used in this study is publicly available on Zenodo (Mojtaba Bezaatpour 2025). This archive includes a Jupyter notebook (VaPOrS.ipynb), a standalone Python script (VaPOrS.py), an input file containing SMILES strings (SMILES.txt), and example output files in both .txt and .csv formats. The repository is licensed under the MIT License and is fully open for use and redistribution under the conditions specified therein.

**Author contributions**

M.B. conceptualized the study and developed the Python code; M.B. and M.R. conducted the functional group analysis, validated the tool against existing data and contributed to data visualization; M.B. prepared the manuscript, and M.D.M. and M.R. reviewed and approved the final version of the manuscript.

**Competing interests**

The authors declare that they have no conflict of interest.

**Acknowledgements**

This project has received funding from the European Research Council under the European Union's Horizon 2020 research and innovation programme under Grant No. 101002728 (ERC Consolidator grant ADAPT) and 101096133 (PAREMPI). This work is also funded by the Research Council of Finland (Grant Nos.: 331207, 336531, 346373, 353836). The AI-based tools were used for language editing to improve the readability of the manuscript.

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
