# Peer review of "VaPOrS v1.0.1: An automated model for functional group detection and property prediction of organic compounds via SMILES notation"

_EGUsphere, 2025_

## Author Comment (AC1)

**Comment on egusphere-2025-2564', Simon O'Meara, 15 Jul 2025:** Bezaatpour et al. present a digital tool (VaPOrS) for estimating pure component saturation vapour pressures, and from this property, the enthalpy of vaporisation. The properties discussed are fundamentally important to understanding aerosols, with significant implications for climate, weather and health. And it is welcome that efforts are being made to further our scientific understanding of this topic. I do hope the authors continue their important work in this area despite my review.

I have a fundamental concern with the submitted paper, which is that it makes an insubstantial contribution to modelling science, making its scientific significance too little to justify publication. Specifically, the referenced UManSysProp tool (Topping et al. 2016) already provides the vapour pressure estimation technique covered by VaPOrS. Then, we ask, do the tools differ significantly in their method to provide these properties? The authors demonstrate in their introduction that there is a variation in method, namely that whilst UManSysProp depends on the OpenBabel package to convert SMILES to SMARTS, which are then parsed, VaPOrS parses the SMILES directly. UManSysProp depends on a self-contained, human-defined, library of SMARTS to identify contributing groups (as described in and around Figure. 3 of Topping et al. 2016), whilst VaPOrS depends on a self-contained, (as far as I understand the paper, human-defined), library of SMILES to identify contributing groups. The Introduction of the paper argues that the VaPOrS method could give better control over pattern-matching logic than is possible in UManSysProp, however I can't see how this is true as both methods rely on a human to provide comprehensive libraries of relevant patterns (SMILES or SMARTS), and so the theoretical maximum degree of control is the same for both methods. Because this issue of insubstantial modelling significance is so important (justifying my rejection for publication) I do not provide further comments on other aspects of the paper at this stage.

**Response:**

We appreciate the reviewer's concern regarding the novelty and significance of our contribution, particularly in relation to the existing UManSysProp tool. In our original manuscript, we deliberately chose not to emphasize direct comparisons with established methods in order to remain neutral and objective, aiming to allow users and modelers to evaluate tools based on their specific needs. However, in light of the reviewer's comment

challenging the merits of VaPOrS relative to UManSysPro, we find it necessary to clarify and emphasize the methodological and practical strengths of our approach to defend its validity and utility. We respectfully submit that VaPOrS provides important advancements that directly address known limitations of UManSysPro, particularly in reading molecular structural information from a complex, general SMILES representation with subsequent estimation of condensational parameters that are critical for secondary organic aerosol (SOA) modeling.

**1. Motivation from Practical Deficiencies in UManSysProp**

While UManSysProp incorporates SMILES notations for vapor pressure estimation, our work originated from repeated, verifiable failures of UManSysProp in correctly identifying necessary functional groups in a wide range of organic species. These failures compromise the integrity of vapor pressure estimation, especially for chemically complex molecules relevant to atmospheric oxidation and SOA formation.

Specifically, we benchmarked UManSysProp and VaPOrS against the original compounds listed in the SIMPOL development paper (Pankow and Asher 2008). We found that UManSysProp failed to identify several functional groups essential for correct vapor pressure computation in some of these molecules (see Table 1). In contrast, VaPOrS correctly computed their vapor pressures in alignment with SIMPOL outputs.

**Table 1.** Benchmark of UManSysProp and VaPOrS against the original SIMPOL compounds from Pankow and Asher (2008). Several functional groups critical for vapor pressure estimation were missed by UManSysProp, resulting in deviations from SIMPOL values. VaPOrS accurately reproduced SIMPOL predictions in all cases. All vapor pressure values are expressed as log p (in atm).

| Compound | SMILES | SIMPOL | UManSysProp | VaPOrS | Experimental |
|---|---|---|---|---|---|
| formamide | C(=O)N | -2.6493 | 1.8307 | -2.6493 | -3.1778 |
| ethyl-formamide | CCNC=O | -3.2014 | 1.1292 | -3.2014 | -3.0353 |
| methyl-formamide | CNC=O | -2.8507 | 1.4799 | -2.8507 | -2.6507 |
| diethyl-formamide | CCN(CC)C=O | -2.0970 | 0.4278 | -2.0970 | -1.8235 |
| dimethyl-formamide | CN(C)C=O | -1.3956 | 1.1292 | -1.3956 | -1.4152 |
| dimethyl-hydroxylamine | CN(C)O | -1.1686 | 1.4799 | -1.1686 | -0.7208 |
| n-butyl-benzoate | CCCCOC(=O)c1ccccc1 | -3.3742 | -2.2884 | -3.3742 | -3.4300 |
| 2-methyl-propyl-benzoate | CC(C)COC(=O)c1ccccc1 | -3.3742 | -2.2884 | -3.3742 | -2.9881 |

| | | | | | |
|---|---|---|---|---|---|
| n-propyl-benzoate | CCCOC(=O)c1ccccc1 | -3.0235 | -1.9377 | -3.0235 | -2.7297 |
| ethyl-benzoate | CCOC(=O)c1ccccc1 | -2.6728 | -1.5870 | -2.6728 | -2.5309 |
| methyl-benzoate | COC(=O)c1ccccc1 | -2.3220 | -1.2363 | -2.3220 | -2.3216 |
| acetic-acid,-phenyl-ester | CC(=O)Oc1ccccc1 | -2.3220 | -2.1088 | -2.3220 | -2.2995 |
| dimethyl-1,2-benzenedicarboxylate | COC(=O)c1ccccc1C(=O)OC | -4.1093 | -1.9377 | -4.1093 | -4.2168 |
| dimethyl-benzene-1,3-dicarboxylate | COC(=O)c1cccc(c1)C(=O)OC | -4.1093 | -1.9377 | -4.1093 | -4.1260 |
| dimethyl-benzene-1,4-dicarboxylate | COC(=O)c1ccc(cc1)C(=O)OC | -4.1093 | -1.9377 | -4.1093 | -4.0839 |
| di-n-butyl-ethanedicarboxylate | CCCCOC(=O)C(=O)OCCCC | -3.4973 | -1.3257 | -3.4973 | -3.3166 |
| diethyl-ethanedicarboxylate | CCOC(=O)C=CC(=O)OCC | -2.7959 | -0.7558 | -2.7959 | -2.5070 |
| ethyl-2-nitropropionate | CCOC(=O)C(C)[N+](=O)[O-] | -3.3118 | -2.4828 | -3.3118 | -2.6618 |
| methyl-2-nitro-propionate | CC(C(=O)OC)[N+](=O)[O-] | -2.9611 | -2.1320 | -2.9611 | -2.6133 |

In a broader evaluation, we assessed 126 primary VOCs provided by the Master Chemical Mechanism (MCM). UManSysProp produced incorrect vapor pressure estimates for at least four compounds (formaldehyde, formic acid, methyl ester, and BCARY) due to inaccurate group detection, while VaPOrS consistently matched the SIMPOL-based expectations (see Table 2).

**Table 2.** Evaluation of four primary VOCs from the MCM database. UManSysProp produced incorrect vapor pressure values due to the misidentification of functional groups, while VaPOrS matched SIMPOL outputs precisely. All vapor pressure values are expressed as log p (in atm).

| Compound | SMILES | SIMPOL | UManSysProp | VaPOrS |
|---|---|---|---|---|
| **formaldehyde** | C=O | 0.6863 | 1.8307 | 0.6863 |
| **formic acid** | OC=O | -1.2397 | 1.8307 | -1.2397 |
| **methyl ester** | COC=O | 0.3942 | 1.4799 | 0.3942 |
| **BCARY** | C/C1=C/CCC(=C)C2CC(C)(C)C2CC\1 | -3.5640 | -3.4531 | -3.5640 |

The implications become even more pronounced when analyzing oxidation products of major VOCs like benzene and α-pinene. For instance, UManSysProp failed to compute correct vapor

pressures for at least 20 α-pinene oxidation products (see Table 3) and 20 benzene oxidation products (see Table 4) due to missed functional groups. VaPOrS, on the other hand, successfully identified these groups and produced accurate results for all cases.

[revised manuscript text omitted]

The challenges identified above are not isolated cases but represent systemic limitations of the existing tool when applied to chemically diverse and rapidly growing databases of atmospheric oxidation products. With new high-resolution mass spectrometry findings continually introducing thousands of new molecules into atmospheric models, a tool like VaPOrS that offers reliable, transparent, scalable, and portable group detection and vapor pressure estimation becomes a valuable contribution to the field.

*2. Methodological Differences and Computational Efficiency*

VaPOrS provides **full control and transparency** over the patterns and matching logic used for functional group identification and vapor pressure calculation directly from SMILES strings, making the process fully auditable and easily modifiable. In contrast, the internal operations of Open Babel used by UManSysPro via its Python wrapper Pybel, are less transparent. Modifying behavior within Open Babel typically requires deep expertise in C++ and SWIG (Simplified Wrapper and Interface Generator), which imposes a steep learning curve on noncore developers to extend or troubleshoot functional group detection [1-3]. Users of UManSysPro have limited visibility into or control over these pattern definitions, complicating diagnosis of missed functional groups or extending detection logic without advanced knowledge of the underlying cheminformatics library.

The reliance on third-party libraries (Open Babel and Pybel) also introduces **computational overhead** in UManSysPro because Pybel converts SMILES strings into internal molecular graph objects before querying functional groups. This step is heavier computationally compared to direct pattern matching on SMILES strings. In large-scale atmospheric secondary organic aerosol modeling scenarios, where thousands of species are processed across thousands of time steps and spatial grid cells, this overhead multiplies substantially, leading to significant decreases in performance. By bypassing the need for Open Babel and Pybel, **VaPOrS can reduce runtime dramatically at atmospheric modeling scales**. For example, with 100 compounds, 10,000 spatial grid cells, and 10,000 time steps, equating to approximately $10^{10}$ vapor pressure calculations, even saving as little as 1 millisecond per call yields an overall runtime reduction on the order of days to weeks of CPU time. This order-of-magnitude speedup exemplifies the practical computational advancement that direct SMILES parsing and pattern matching, as implemented in VaPOrS, can provide over UManSysPro's Pybel-dependent approach.

Even though VaPOrS's Pybel-free Python code performs much faster than UManSysPro's Pybel-based implementation, it remains subject to **Python interpreter overhead** and suboptimal scaling compared to compiled languages. Fortran, for example, is the language of choice for most leading-edge high-performance simulation codes such as PALM [4] and ADCHEM [5], due to its superior efficiency in large-scale numerical computations and direct memory management [6]. Direct comparisons show Fortran can be hundreds of times faster than pure Python for the same algorithm. For example, a realistic computational task took 66 seconds in Fortran versus 33,900 seconds (9.5 hours) in Python, a factor of 500 difference [7]. Fortran manages memory more efficiently, reducing runtime and minimizing memory exhaustion risks compared to Python's dynamic and object-heavy environment. Modern Fortran, with support for OpenMP and MPI, is well suited for parallelizing across multi-core CPUs and HPC (high-performance computing) clusters, ideal for the demanding grids and time steps of atmospheric SOA modeling. Compiled Fortran code runs consistently across platforms, avoiding variability caused by the Python interpreter and dynamic dependency issues. Regarding this, UManSysPro's dependency on Open Babel/Pybel creates challenges for

portability and maintainability. Open Babel is a large C++ library with Python bindings, and porting its entire dependency chain to other programming languages, especially Fortran, is impractical and cumbersome. The typical solution, such as writing complex wrappers or using foreign function interfaces, is technically challenging and fragile. Fortran's limited interoperability with C++ libraries without extensive tooling contrasts sharply with VaPOrS's algorithmic approach, which is based solely on standard string operations and pattern matching, making it **straightforward to port to Fortran**. Furthermore, calling UManSysPro, which uses Python, from the above-mentioned Fortran-based models for vapor pressure calculations at every time step and grid cell for many compounds introduces a **mixed-language dependency that can incur significant runtime overhead**. This overhead arises from interpreter invocation and data marshalling between Fortran and Python, **cumulatively adding up to the slowdown** already caused by Pybel.

If the vapor pressure calculation code is ported natively into Fortran, **as VaPOrS's algorithmic simplicity allows straightforward translation to Fortran**, these penalties vanish. Fortran's compiled nature allows **massively faster execution**, seamless integration, and optimized parallelism, critical for atmospheric scale modeling. Pure numerical and string logic implemented in Fortran will maximize CPU throughput and scale smoothly across large model domains without bottlenecks imposed by language design. Therefore, VaPOrS's practical and potential advantages in reducing runtime can be summarized as follows:

- **Avoiding unnecessary computations** by bypassing Pybel's SMILES-to-graph conversion and using direct pattern matching.
- **Eliminating Python–Fortran interface overhead** by avoiding mixed-language calls during large-scale simulations.
- **Allowing pure Fortran translation**, enabling much faster native execution compared to Python.

*3. Execution Conveniency*

Running UManSysPro on supercomputers (high-performance computing, HPC) can lead to several complainable problems, especially in large-scale modeling workloads typical of atmospheric science. Supercomputers frequently run custom or stripped-down Linux environments. Installing complex dependencies like Open Babel and its Python wrappers

(Pybel, SWIG bindings) is often problematic. Users may need special compilation steps, manual resolution of C++/Python/Swig compatibility, or may encounter issues with missing or incompatible shared libraries. Open Babel/Pybel and all dependent Python packages must be installed and maintained across all compute nodes. Minor discrepancies in versions or build environments between nodes can cause errors that are very hard to track in distributed jobs, leading to wasted supercomputing allocations. Furthermore, some supercomputer environments restrict the installation of non-standard libraries or require jobs to run only using centrally installed software. Getting approval or support for Open Babel/Pybel can be challenging, especially when the core libraries are actively developed with potentially breaking changes [8-10]. These factors are widely acknowledged as major barriers to deploying Python-based, C++-linked scientific tools like UManSysPro on supercomputers compared to simpler, compiled, single-language scientific code. These issues often motivate writing more portable, lightweight, and easily compilable tools for large HPC applications, or at least favoring solutions that can be run natively in Fortran on all target architectures without Python or complex bindings.

*Conclusion*

While both UManSysProp and VaPOrS nominally implement the SIMPOL method, the pathway to functional group identification and vapor pressure computation is substantially different. VaPOrS's direct SMILES parsing avoids known limitations of Pybel-based systems, offers a more flexible and extensible framework, and has demonstrably outperformed UManSysProp across multiple atmospheric datasets, including benchmark compounds, MCM species, oxidation products, and autooxidation products.

We hope this detailed explanation addresses the reviewer's concern regarding the significance and novelty of our approach and demonstrates that VaPOrS represents a robust advancement in modeling vapor pressure for atmospheric applications.

---

## Author Response (AR1)

**Response to Reviewers**

We sincerely thank the Editor and the reviewers for their valuable time and insightful comments on our manuscript. We appreciate the constructive feedback, which has helped us significantly improve the quality and clarity of the paper. All comments have been carefully addressed, and the manuscript has been revised accordingly. Below, we provide a point-by-point response to each comment raised by the reviewers. All changes in the manuscript are highlighted in yellow color.
* * *
**Reviewer 1**

**Comment:** Many thanks for your comprehensive reply. I appreciate the time and effort that has been dedicated to both the original paper and the response. The evidence presented in the response does demonstrate the significance of VaPOrS, in contrast to my evaluation in my original review. I will contact the editor to ask whether another review, in light of the response, is allowed/wanted. If permitted, I will suggest the supplied response be included as a major revision to the original paper.

**Response:** *We sincerely thank you for your constructive follow-up and for reconsidering your initial evaluation in light of our response. We greatly value the time and effort you have devoted to reviewing our work and for acknowledging the significance of VaPOrS in the revised context. Your thoughtful feedback has been instrumental in strengthening the manuscript, and we are grateful for your support in moving this work forward. We included the supplied response as a new subsection of the 'Results and discussion' (see subsection 4.3) in the revised manuscript. Also, we allocated the last paragraphs of the 'Introduction' section to this matter, as follows:*

Despite the usefulness of existing tools such as UManSysProp (Topping et al. 2016) for vapor pressure estimation, they show limitations in correctly identifying functional groups in a range of organic molecules. These shortcomings can lead to significant deviations in predicted vapor pressures, particularly for multifunctional species relevant to atmospheric oxidation and SOA formation. Our observations of such discrepancies provided the main motivation for this work. To address these deficiencies, we developed a Python-based computational framework named VaPOrS (**Va**por **P**ressure in **Or**ganics via **S**MILES) to process SMILES (Simplified Molecular Input Line Entry System) notation of VOCs, automatically identify functional groups, and apply group-contribution methods for property estimation.

The core innovation of VaPOrS lies in its self-contained SMILES parsing and group recognition algorithm, which eliminates reliance on external cheminformatics libraries such as OpenBabel. Instead of depending on SMARTS-based pattern matching, VaPOrS explicitly searches for all possible patterns of each functional group directly from the SMILES string, ensuring full control over the detection logic. This approach makes the tool both transparent and adaptable, enabling straightforward extension to new group definitions and methods without external dependencies. In its current version, VaPOrS implements the SIMPOL method by detecting 30 functional groups required for estimating saturation vapor pressure and enthalpy of vaporization. However, this is only the first demonstration of the framework: the same group recognition functions can be applied to other parameterization schemes (e.g., group additivity, volatility basis set (VBS) models, partition coefficients, Henry's law constants), making VaPOrS a general platform for group-contribution modeling rather than a tool restricted to vapor pressure prediction. Therefore, the development of VaPOrS addresses several challenges:

1. Automated and auditable functional group detection: Eliminating manual identification, reducing error potential, and providing full transparency in detection logic.

2. Computational efficiency: By bypassing Pybel's SMILES-to-graph conversions, VaPOrS reduces overhead, enabling much faster execution for large-scale atmospheric simulations involving thousands of compounds across many steps and grid cells.

3. Scalability and flexibility: Capable of processing thousands of SMILES strings within seconds, with design features that make it portable to high-performance computing (HPC) environments and easily translatable to compiled languages such as Fortran for integration into large-scale chemical transport and climate models.
* * ** * *
**Reviewer 2**

This manuscript introduces VaPOrS v1.0.1, a Python-based tool developed to estimate saturation vapor pressure and enthalpy of vaporization for organic compounds using their SMILES representations. A key feature of VaPOrS is its built-in capability to detect functional groups directly from SMILES strings, eliminating the need for external cheminformatics

libraries and manual SMARTS definitions. The tool relies on the SIMPOL group contribution method developed by Pankow and Asher for its vapor pressure predictions and, at the moment is able to recognize only the 30 functional groups needed to apply this method.

The authors validated VaPOrS against the original SIMPOL dataset and demonstrated perfect agreement between the two approaches. Further testing on an external dataset (i.e., MCM database) showed strong correlation with manually derived SIMPOL predictions and other established models such as EVAPORATION and Nanoolal. The methodology is sound, and the tool appears robust and computationally efficient and has potential to be further expanded.

I recommend the manuscript for publication after revisions are made to address the concerns outlined below.

**Comment:** I think the main contribution of this work is the development of the functions to detect functional groups. This is done in a efficient way directly in the tool without relying on external libraries. I know how frustrating can be installing and setting up dependencies between different libraries and tools and I value a self-contained tool that can be adapted to include different methods. So, I think the strongest point of the paper is the SMILES parser and groups identification. The authors present a first implementation of the SIMPOL methods and highlight that the tool can be expanded to include more group contribution methods to predict saturation vapour pressure. However, this feels somewhat restrictive, as the group recognition framework developed in VaPOrS is broadly applicable and could be adapted to predict a wider range of physicochemical properties (e.g., partition coefficients) and not only vapor pressure. I think this point should be stressed more in the manuscript and the tool should be presented as a general tool for SMILES parsing and group contribution method application. Conversely the authors mainly focus on describing the SIMPOL implementation for the prediction of VP. This is an established method developed by other scientists. At a first reading it appears the VaPOrS just apply the SIMPOL method without apporting any contribution, thus I understand the comments of the first reviewer. The main contribution of the paper is the automatization of the fragments recognition in an efficient way and I think this should be stressed more.

Related to this, since the real novelty are the SMILES parsing functions, I think a substantial validation of the group recognition method is missing in the paper. Section 3.4.1 (MCM data) briefly describes as the SMILES parsing functions have been tested on 126 external compounds. I think this should be one of the main sections of the paper demonstrating that the functions are able to correctly recognize the functional groups needed by SIMPOL (or any

other group contribution method implemented) on an external dataset. The authors have provided supplementary material in response to a previous reviewer's comment, comparing their approach with UManSysProp and highlighting cases where UManSysProp fails to correctly identify certain groups, leading to inaccurate predictions. This comparison is highly relevant and should be integrated into the main text to underscore the robustness and reliability of VaPOrS. The authors criticize the SMART pattern recognition in OpenBabel, so a comparison between fragments identified by VaPOrS and fragments identified by OpenBabel should be included to highlight the strength of VaPOrS related to OpenBabel and justify the development of ad-hoc functions in a new method.

**Response:** *We sincerely thank the reviewer for the constructive and thoughtful comments, which helped us to better highlight the true novelty and contributions of our work. We agree with the reviewer that the core innovation of our study lies in the development of efficient, self-contained functions for functional group recognition directly from SMILES, without relying on external libraries. We revised the manuscript to emphasize this point more clearly. In particular, we framed VaPOrS not only as a tool implementing the SIMPOL method, but more broadly as a general framework for SMILES parsing and group contribution method applications, adaptable to the prediction of a wider range of physicochemical properties such as partition coefficients, Henry's law constants, and activity coefficients. We acknowledge that in the current version, the manuscript places more focus on the SIMPOL implementation, which may give the impression that the work simply replicates an established method. In the revised version, we shifted the emphasis toward the novelty of the group recognition framework and its versatility, presenting the SIMPOL implementation as a case study to demonstrate the functionality and accuracy of the approach. We addressed this in different sections of the revised manuscripts as follows:*

*In the Abstract:*

[revised manuscript text omitted]

*Regarding validation, we fully agree with the reviewer that a substantial demonstration of the robustness of the group recognition method is essential. To address this, we reorganized the sections of the revised manuscript, allocating a main section to validation of VaPOrS, where we evaluated its accuracy in predicting the saturation vapor pressures and enthalpy of vaporization of 224 organic compounds listed in the SIMPOL-pertained study (see section 3). Moreover, we expanded the main sections of the 'Results and discussion', demonstrating the capability of VaPOrS to correctly recognize the functional groups needed by SIMPOL on external datasets provided by MCM and autoAPRAM-fw (see sections 4.1 and 4.2). Finally, we integrated into the main text the comparison with UManSysProp, currently included in the supplementary material, as it clearly illustrates how VaPOrS overcomes the limitations of existing tools, specifically those relying on OpenBabel and SMARTS-based approaches, in group recognition and vapor pressure prediction.*

*We believe these revisions clarify the originality of the work, underscore the robustness of the developed framework, and address the reviewer's concern that the manuscript currently underemphasizes its main contributions.*

**Comment:** Page 5, line 4, […tools like VaPOrS, enabling…] VaPOrS acronym has not been established yet.

**Response:** *We thank the reviewer for pointing this out. In the revised manuscript, we have now introduced the full name of the tool at its first occurrence as VaPOrS (Vapor Pressure of Organics via SMILES), after which the acronym is used consistently (see line 12, page 5).*

**Comment:** Page 7, lines 19-21 [In particular, the SMILES string must begin…], the authors write MUST BEGIN implying that the SMILES need to be provided in a specific way. A SMILES for a chemical can be written in many different variation (e.g, canonical vs kekulized). To be valid, the tool must be able to recognize functional groups even for different variation of the same SMILES. Given that SMILES syntax can vary depending on generation method or canonicalization, it is important to demonstrate that the tool yields consistent fragment counts regardless of input variation. This would strengthen confidence in the robustness of the group recognition algorithm and its suitability for large-scale automated analyses. The manuscript should also clarify whether VaPOrS includes a SMILES standardization step prior to functional group parsing. Standardization is essential to ensure reproducibility in fragment recognition.

**Response:** *We thank the reviewer for highlighting the importance of SMILES variation and standardization. At the current stage, VaPOrS operates using canonical SMILES as input. This choice was made because canonical SMILES are widely used in chemical databases and ensure a unique representation of each molecule, which simplifies functional group detection and validation. We acknowledge that the lack of an internal standardization step limits flexibility when handling alternative SMILES notations. We plan to extend future versions of VaPOrS to include this functionality. This will require additional development time, as the current algorithm was designed and validated specifically with canonical SMILES. To clarify this point for readers, we have revised the manuscript to explicitly state that VaPOrS presently requires canonical SMILES as input. This has been brought in the 'Methodology' section as follows:*

At the current stage, VaPOrS operates using canonical SMILES as input. This choice was made because canonical SMILES are the most common representation in chemical databases and ensure a unique representation of each molecule, which simplifies functional group detection and validation. Although alternative SMILES forms (e.g., kekulized or non-canonical variations) can represent the same molecule, these are not yet supported in the present version. To maintain consistency, users should therefore provide canonical SMILES when running VaPOrS.

**Comment:** Furthermore, the manuscript should address how VaPOrS handles tautomeric variability in SMILES representations. Tautomers are chemically equivalent but structurally distinct forms of the same chemical that can be encoded differently. This variability can significantly impact functional group recognition and, consequently, the accuracy of property predictions. It is unclear whether the authors have tested the tool's consistency across different tautomeric forms of the same compound. I recommend including a discussion on this issue and, if not already performed, conducting a validation study to assess whether VaPOrS yields consistent fragment counts and predictions across tautomeric variants.

**Response:** *We thank the reviewer for raising the issue of tautomeric variability in SMILES representations. As noted, tautomerism can lead to structurally distinct encodings of the same compound, which in turn may alter the number and type of functional groups identified. VaPOrS is designed to parse SMILES and count these functional groups exactly as they are presented. Therefore, any variability in the predicted properties between tautomeric SMILES does not arise from VaPOrS itself, but rather from the group-contribution framework applied (in this case, SIMPOL). Since functional group–based methods such as SIMPOL assign unique parameters to each group (e.g., C=C, OH, and C=O), different tautomeric inputs naturally yield different parameterizations. Currently, SIMPOL and similar models do not provide distinct parameter sets for tautomeric forms, leaving this ambiguity unresolved. Should future studies propose a robust strategy, such as allocating distinct parameters for tautomeric groups, this could be readily implemented in VaPOrS. For now, VaPOrS follows the input SMILES representation provided by the user or database and detects C=C, OH, and C=O as distinct functional groups, while acknowledging that tautomerism is a limitation of group-contribution methods rather than of VaPOrS itself. We have now added a detailed discussion of tautomerism as a limitation of group-contribution approaches such as SIMPOL in the revised manuscript. This has been brought in the 'Results and discussion' section as follows:*

Another important limitation relates to tautomerism, in which a compound can exist in multiple chemically equivalent but structurally distinct forms, for example keto-enol tautomerism. These forms contain different functional groups. For instance, a keto tautomer may contain a carbonyl group (C=O), while its enol counterpart may contain one hydroxyl (OH) and one carbon-carbon double bond (C=C). Since functional group-based methods like SIMPOL assign unique parameters to each group, different tautomeric SMILES representations of the same compound can yield different vapor pressure predictions, despite the compound having a single experimentally measurable vapor pressure. VaPOrS does not attempt to canonicalize or

normalize tautomeric forms; it faithfully parses and counts groups in the SMILES provided by the user or database. Thus, this ambiguity arises from the group-contribution framework itself rather than from VaPOrS. In practice, most studies adopt a convention of selecting the thermodynamically more stable tautomer, commonly the keto form in the gas phase, as the reference structure, although this is not universally standardized. Should future investigations establish robust strategies for handling tautomerism, for example by developing distinct parameters for different tautomeric states, such improvements could be readily integrated into VaPOrS. Addressing these limitations, whether through correction terms, hybridization with other frameworks, or leveraging data-driven approaches such as machine learning, represents an important avenue for future development.

**Comment:** Page 30, lines 14-17 [Many data points are clustered close to this line…], this is subjective comments. A more objective description would consider some metric like the R2 or the RMSE. Please provide some quantitative metrics to describe your correlation.

**Response:** *We appreciate the reviewer's suggestion. In the revised manuscript, we have replaced the subjective description with quantitative performance metrics (RMSE and R²) to objectively evaluate the accuracy of the VaPOrS predictions. Specifically, for log P at 333.15 K (Figure 4), we report RMSE = 0.4232 and R² = 0.9648. Across six temperatures (Figure 5), the results yield RMSE = 0.5556 and R² = 0.9570. For vaporization enthalpies at 333.15 K (Figure 6), the comparison gives RMSE = 14.4740 and R² = 0.6146. These values are now included in the Results section to provide a clear, quantitative assessment of model performance.*

**Comment:** Page 32, lines 6-10 [These discrepancies could be attributed to the structural complexity…] this paragraph concern the applicability domain of the model. I know the SIMPOL model has not been developed by these authors, but could VaPOrS provide an applicability domain? Maybe something related to the groups count? For instance, does the presence of certain group together of the presence of too many instance of the same fragment result in more uncertain prediction?

**Response:** *We thank the reviewer for this insightful comment regarding the applicability domain of the SIMPOL model and the potential role of VaPOrS in this respect. While it is true*

*that SIMPOL was not originally designed with an explicit applicability domain, the strength of VaPOrS lies in its ability to decompose molecular structures into functional groups and quantify them systematically. This feature could indeed be leveraged to flag molecules where the presence of multiple instances of the same fragment or unusual combinations of groups may result in higher prediction uncertainty. For example, compounds containing a large number of hydroxyl or carboxyl groups, or rare functionalities such as peroxides, tend to show greater discrepancies from experimental values. We have added a discussion in the manuscript to clarify that although VaPOrS itself does not yet provide a formal applicability domain, it could serve as a diagnostic tool by identifying structural features that are likely to fall outside the reliable range of SIMPOL predictions. This has been brought in the 'Results and discussion' section as follows:*

At the same time, it is important to recognize the limitations of group contribution approaches such as SIMPOL, which underpins this first implementation of VaPOrS. While reliable for many compounds, predictive accuracy declines as molecular complexity increases. Highly functionalized molecules may exhibit non-additive effects, such as steric hindrance, intramolecular hydrogen bonding, or electronic interactions, that are not captured by simple group summation rules. Previous studies have shown that such effects can dampen or amplify volatility changes in ways not accounted for by group contribution approaches.

.

.

.

Addressing these limitations, whether through correction terms, hybridization with other frameworks, or leveraging data-driven approaches such as machine learning, represents an important avenue for future development.

**Comment:** Figure 7, Antoine and SIMPOL methods seem to give a good agreement. However, there are instances where the two methods seem far from the experimental line (Decanedioic acid, Hexanamide, Diethyl-peroxide). Please comment.

**Response:** *We thank the reviewer for this valuable observation. We agree that while both the Antoine and SIMPOL methods generally show good agreement with experimental data, there*

*are notable deviations for certain compounds. As shown in Figure 7, examples include Decanedioic acid, which contains two carboxyl groups, Hexanamide, where strong hydrogen-bonding interactions are important, and Diethyl-peroxide, which contains an uncommon peroxide group. These structural features make intramolecular interactions more significant and less amenable to simple group additivity approaches, which likely explains the deviations. Such cases highlight the limitations of SIMPOL method when applied to structurally complex or less common functionalities, and it also points to the potential of VaPOrS to help identify molecules that fall outside the reliable applicability domain of the SIMPOL framework. We added a discussion on this deviation in the manuscript, as follows:*

Although the SIMPOL model does not explicitly define an applicability domain, the structure-based framework of VaPOrS offers an opportunity to explore this aspect. Since the code identifies and counts functional groups for each molecule, it can be used to highlight cases where predictions may be less reliable. For instance, compounds such as Decanedioic acid, which contains two carboxyl groups, or Hexanamide, where strong hydrogen-bonding interactions may play a role, or Diethyl-peroxide, which contains a relatively uncommon peroxide group show noticeable deviations from experimental values (See Figure 7). These examples suggest that compounds with an unusually high number of hydroxyl or carboxyl groups, or those containing less common fragments such as peroxides, often exhibit larger deviations from experimental vapor pressures. Likewise, the co-occurrence of multiple reactive groups within the same molecule (e.g., carbonyl–peroxide combinations) may introduce additional uncertainties. While a systematic applicability domain analysis is beyond the scope of this study, we note that VaPOrS could be extended to provide such functionality, thereby guiding users in assessing the reliability of SIMPOL predictions for structurally complex molecules.

**Comment:** Figure 11, 3D graphs look cool on computer screen in interactive applications. When on paper are kind of hard to read. For example, I cannot see the depth on one of the axis. I see that the information on the Mass can be interesting, perhaps a 2D correlation between Mass and the groups count would be better for a printed version of the paper.

**Response:** *We appreciate your comment regarding Figure 11. We agree that 3D plots can sometimes be challenging to interpret in printed form. However, we believe the 3D*

*representation provides valuable information by simultaneously illustrating the relationships among Mass, functional group count, and comparing vapor pressures, which would be difficult to capture in a purely 2D format. To enhance clarity and improve readability, we have therefore kept the 3D plot but also added an additional 2D plot (see Figure 12) showing the correlation between Mass and important features, as you suggested. This provides a clearer representation in print while retaining the multidimensional perspective of the original figure.*

*We revised the manuscript accordingly. Using VaPOrS, the benzene-derived oxidation and autoxidation products were further classified into volatility categories based on their effective saturation mass concentration ($C^*$, µg m$^{-3}$) in the newly added Figure 12. These include ultra-low-volatility organic compounds (ULVOC, $C^* \leq 3\times10^{-9}$ µg m$^{-3}$), extremely low-volatility organic compounds (ELVOC, $3\times10^{-9} < C^* \leq 3\times10^{-5}$ µg m$^{-3}$), low-volatility organic compounds (LVOC, $3\times10^{-5} < C^* \leq 3\times10^{-1}$ µg m$^{-3}$), semi-volatile organic compounds (SVOC, $3\times10^{-1} < C^* \leq 3\times10^{2}$ µg m$^{-3}$), and intermediate-volatility organic compounds (IVOC, $3\times10^{2} < C^* \leq 3\times10^{6}$ µg m$^{-3}$). In this figure, we present the 2D relationship between molar mass and effective saturation concentration for these products.*

---

## Author Response (AR2)

**Response to Reviewers**

We sincerely thank the Editor and the reviewers for their valuable time and insightful comments on our manuscript. We appreciate the constructive feedback, which has helped us significantly improve the quality and clarity of the paper. All comments have been carefully addressed, and the manuscript has been revised accordingly. Below, we provide a point-by-point response to each comment raised by the reviewers. All changes in the manuscript are highlighted in yellow color.
* * *
**Reviewer 1**

Bezaatpour et al. (2025) have provided a very valuable paper, and associated tool, to the environmental science community. The method and validation is very comprehensive, and the paper is thoughtful in exploring the limitations of the presented tool and its implemented estimation method. I recommend publication following the minor revisions suggested below.

Introduction

**Comment:** Please state somewhere the type of saturation vapour pressure being estimated, including the Raoult effect and the phase, e.g.: 'pure component liquid (or sub-cooled liquid) saturation vapour pressure'. (where pure component deals with the Raoult effect and liquid (or sub-cooled liquid) deals with the phase.

**Response:** *In the revised manuscript, we stated that the model calculates the pure-component liquid (or sub-cooled liquid) saturation vapor pressure, which corresponds to the vapor pressure of a compound in its pure state (without accounting for Raoult's law mixing effects) and in the liquid or sub-cooled liquid phase, as follows:*

In its current version, VaPOrS implements the SIMPOL method by detecting 30 functional groups required for estimating pure-component (sub-cooled) liquid saturation vapor pressure and the corresponding enthalpy of vaporization, without accounting for Raoult's law effects.

**Comment:** Line 6 of Page 5, could you reference the relevant results section of your paper when discussing the UManSysProp inaccuracies? For readers interested in this inaccuracy, doing so will save them time searching for related information.

**Response:** *In the revised manuscript, we added a reference to the relevant section (i.e., 4.3) discussing the UManSysProp inaccuracies.*

Section 3

**Comment:** Line 8 Page 30 and Line 14 Page 31 and Line 14 Page 33

Whereas the authors reference the original SIMPOL paper for the source of measured vapour pressures in Fig. 4. It's unclear where the measurements for Fig. 5, Fig. 6 and Fig. 7 come from, please provide reference(s).

**Response:** *The measured vapor pressures used for validation in Figures 5–7 were obtained from the same experimental datasets reported in the original SIMPOL study by Pankow and Asher (2008). In the revised manuscript, we clarified the sources of the measured vapor pressure data. The corresponding reference has now been explicitly cited in the figure captions and in the text.*

Results

**Comment:** Line 7 Page 36

In terms of what is available to users from the internet, the current MCM website (https://www.mcm.york.ac.uk/MCM/ accessed October 15th 2025) does not provide a file containing SMILES of all MCM species. This has been confirmed with Killian Murphy and Andrew Rickard at University of York. If it helps the authors direct readers to such a file, for the current version of MCM, v3.3.1, a file is stored here, with no plans to remove it: https://github.com/simonom/PyCHAM/blob/master/PyCHAM/prop_store/mcm_v3p3p1_all_species.xml.

I see (code availability section) that the MCM v3.3.1 SMILES may also be available in the zenodo repository. If so, please state explicitly in the code availability whether this is the case. Either way, please make clearer at line 7 page 36 where readers can obtain the MCM v3.3.1 SMILES.

**Response:** *In the revised manuscript, we stated that the current version of the MCM (v3.3.1) can be obtained as the file mcm_3-3-1_species_complete.tsv from the MCM archive page ( https://www.mcm.york.ac.uk/MCM/about/archive)*

Code availability

**Comment:** Could a URL or DOI be provided for the relevant zenodo repository?

**Response:** *In the revised manuscript, we added the following DOI to the Code availability section:* https://doi.org/10.5281/ZENODO.15222175
* * *
**Reviewer 2**

The manuscript has improved substantially and is now ready for publication. I'm pleased to see that the main points raised by the reviewers have been addressed. The focus has been shifted toward highlighting the novelty and versatility of the group recognition framework, with the SIMPOL implementation presented as a case study demonstrating its potential applications. The addition of the new tables provides validation data that illustrate the framework's superior performance compared to similar tools.

In my opinion, the fact that the tool does not perform SMILES standardization and relies on the user to provide correct input SMILES remains a limitation. However, this limitation is transparently acknowledged in the manuscript. I'm glad to see that the authors have plan for address this in the future.

**Comment:** I think the conclusions of the paper feel heavily written by AI (most of the acronyms are spelled out even if were explained before, please address this). I'd rather a more "human-like" tone.

**Response:** *We appreciate the reviewer's comment. Initially, we had expanded acronyms in the Conclusions section to ensure clarity for readers who might focus only on this section. However, based on the reviewer's suggestion, we have now removed these expansions to make the conclusions read more naturally while retaining all key information.*

**Comment:** Finally, on page 7, line 15, please remove the word "unique." Canonical SMILES do not represent a unique molecular representation.

**Response:** *We thank the reviewer for this comment. We agree that the term "unique" is not accurate in this context, as canonical SMILES do not guarantee a single molecular representation. Accordingly, we have removed the word "unique" from Page 7, Line 15 in the revised manuscript.*
* * *
**Reviewer 3**

**Comment:** wherever an estimated vapour pressure is provided, the corresponding temperature should be stated in the main text and Figure/Table caption, for example the Figures in Section 4.2 and Tables in Section 4.3. Further clarity is needed when vapour pressures from 'SIMPOL' or the 'SIMPOL method' are provided. If these have been calculated manually, this should be stated in each sub-section of results, including in Sections 3.1 and 4.3. Or, if these have been obtained from the original Pankow and Asher paper (e.g. using the Antoine coefficients in their supplement), this should be stated (including where in the paper they come from). If there is an alternative source, then this should be stated. Otherwise the assumption is that values have come directly from the original paper, which could mean a different interpretation than for manual generation.

**Response:** *We thank the reviewer for this suggestion. In the revised manuscript, all estimated vapor pressures now explicitly state the corresponding temperature in the main text as well as in figure and table. Furthermore, we clarified the source of all vapor pressures labeled as "SIMPOL", "VaPOrS", "Experimental" and "UManSysProp". These updates ensure readers can distinguish between calculated and literature-derived values throughout the relevant sections.*

**Comment:** There is a degree of circularity that arises from the same authors writing the matching patterns for the VaPOrS tool and manually estimating vapour pressures, when both are based on their interpretation of the SIMPOL rules. Ideally the VaPOrS estimates would be compared against independent estimates, e.g. from the original SIMPOL paper, though unfortunately that paper does not seem to supply the estimates in tabulated form. But there is some interpretation of the SIMPOL rules needed, and some of the chemicals presented in Section 4.3 exemplify the point. E.g. dimethyl-hydroxylamine is strictly a hydroxylamine, not an amine, though my calculations show that the authors have included the secondary amine group to achieve the manually estimated (SIMPOL) vapour pressure. And to be fair, Table 1C of the Pankow and Asher paper lists this molecule as an amine, suggesting a consistency with the intended meaning of this group. The same molecule includes a hydroxyl group bonded to a nitrogen atom, but both Table 5 of the Pankow and Asher paper and Section 2.1.5 of the paper under review refer to the hydroxyl group to be considered for the group contribution as an alkyl hydroxyl, which my understanding means the hydroxyl group bonded to a non-aromatic carbon atom. Arguably

because the bond is to nitrogen rather than carbon the hydroxyl group should not contribute to the estimated vapour pressure in this case, but my calculations show that the authors have included it for their manual SIMPOL estimate. I think the authors need to acknowledge at least these points of interpretation in the paper, plus any other points where interpretation has been needed for the SIMPOL rules.

**Response:** *We appreciate the reviewer's insightful observation regarding the interpretation of the SIMPOL functional group definitions, particularly in relation to compounds such as dimethyl-hydroxylamine. We acknowledge that certain group classifications required interpretative judgment since the original SIMPOL paper did not provide tabulated vapor pressure values or exhaustive group assignments for all compounds. Our classification choices were guided strictly by the published definitions and data representations in Pankow and Asher (2008). Specifically, while the SIMPOL framework defines alkyl hydroxyls as hydroxyl groups bonded to non-aromatic carbon atoms, our observation of Figures 7(a) and 13 indicated that the hydroxyl group in dimethyl-hydroxylamine contributed to both vapor pressure and enthalpy of vaporization. Since the numerical data were not available despite our direct request from the original authors, our interpretation was necessarily based on visual analysis of the published figures. We therefore adopted this interpretation in VaPOrS to remain consistent with the apparent treatment in the SIMPOL study. Importantly, VaPOrS is designed to be flexible and can be updated to incorporate any future clarifications or refinements provided by the SIMPOL authors or by users of the tool. We addressed this matter in the revised manuscript as follows:*

It is worth mentioning that in compounds such as dimethyl-hydroxylamine, the initial classification considered only the amine functional group, excluding the hydroxyl, as the SIMPOL definition specifies alkyl hydroxyls as hydroxyl groups bonded to non-aromatic carbon atoms. However, careful observation of Figures 7(a) and 13 in (Pankow and Asher 2008) revealed that the hydroxyl group appears to have been included in the calculation of both vapor pressure and enthalpy of vaporization. Although the numerical values were unavailable, this observation led us to infer that the phrase "non-aromatic carbon" may have been used to distinguish the behavior of hydroxyl groups attached to aromatic rings rather than to strictly define the atom of attachment. Accordingly, VaPOrS adopts this interpretation to align with the apparent implementation in SIMPOL. The tool remains adaptable and can be updated should new clarifications or refinements to the SIMPOL framework become available.